

# Quantitatively assessing mekosuchine crocodile locomotion by geometric morphometric and finite element analysis of the forelimb

Michael D. Stein[1], Suzanne J. Hand[1], Michael Archer[1], Stephen Wroe[2] and Laura A.B. Wilson[1]

[1] PANGEA Research Centre, School of Biological, Earth and Environmental Sciences, University of New South Wales, Sydney, New South Wales, Australia
[2] Function, Evolution and Anatomy Research Laboratory, School of Environmental and Rural Sciences, University of New England, Armidale, New South Wales, Australia

Corresponding author
Michael D. Stein,
michael.stein@unswalumni.com

## ABSTRACT

Morphological shifts observed in the fossil record of a lineage potentially indicate concomitant shifts in ecology of that lineage. Mekosuchine crocodiles of Cenozoic Australia display departures from the typical eusuchian body-plan both in the cranium and postcranium. Previous qualitative studies have suggested that these crocodiles had a more terrestrial habitus than extant crocodylians, yet the capacity of mekosuchine locomotion remains to be tested. Limb bone shape, such as diaphyseal cross-section and curvature, has been related to habitual use and locomotory function across a wide variety of taxa. Available specimens of mekosuchine limbs, primarily humeri, are distinctly columnar compared with those of extant crocodylians. Here we apply a quantitative approach to biomechanics in mekosuchine taxa using both geomorphic morphometric and finite element methods to measure bone shape and estimate locomotory stresses in a comparative context. Our results show mekosuchines appear to diverge from extant semi-aquatic saltwater and freshwater crocodiles in cross-sectional geometry of the diaphysis and generate different structural stresses between models that simulate sprawling and high-walk gaits. The extant crocodylians display generally rounded cross-sectional diaphyseal outlines, which may provide preliminary indication of resistance to torsional loads that predominate during sprawling gait, whereas mekosuchine humeri appear to vary between a series of elliptical outlines. Mekosuchine structural stresses are comparatively lower than those of the extant crocodylians and reduce under high-walk gait in some instances. This appears to be a function of bending moments induced by differing configurations of diaphyseal curvature. Additionally, the neutral axis of structural stresses is differently oriented in mekosuchines. This suggests a shift in the focus of biomechanical optimisation, from torsional to axial loadings. Our results lend quantitative support to the terrestrial habitus hypothesis in so far as they suggest that mekosuchine humeri occupied a different morphospace than that associated with the semi-aquatic habit. The exact adaptational trajectory of mekosuchines, however, remains to be fully quantified. Novel forms appear to emerge among mekosuchines during the late Cenozoic. Their adaptational function is considered here; possible applications include navigation of uneven terrain and burrowing.

## INTRODUCTION

The shape and biomechanical properties of long bones often reflect locomotor function, not only across evolutionary time spans but within an individual's lifetime as living bone tissue responds to the stresses commonly encountered in life (e.g., *Ruff, Holt & Trinkaus, 2006*; *Shaw & Stock, 2009*). Variation in shape can be quantified using well established geometric morphometric techniques which are increasingly being integrated with a number of non-destructive, computer-based engineering techniques that predict magnitude and distributions of stress, strain, and deformation in structures (*Adams, Rohlf & Slice, 2004*; *O'Higgins & Milne, 2012*; *Piras et al., 2012*; *Polly et al., 2016*; *Wroe et al., 2018*). Finite element analysis (FEA) in particular, is a powerful computational approach that has been used by engineers for decades to predict mechanical behaviour in man-made structures. Over the last ten years, FEA has been increasingly applied to biological structures (*Rayfield, 2007*; *Wroe et al., 2007*; *Wroe, 2008*; *Wroe et al., 2008*; *Wroe, 2010*; *Walmsley et al., 2013b*). In the field of palaeontology, FEA has most commonly been used to predict the capacity of crania and mandibles to sustain measured or predicted forces generated through biting or killing behaviours, and to examine questions surrounding feeding ecology (*Rohlf & Marcus, 1993*; *Taylor et al., 1996*; *Richmond et al., 2005*; *Rayfield, 2007*; *Wroe et al., 2007*; *Wroe, 2008*; *Wroe et al., 2008*; *Wroe, 2010*; *Parr et al., 2012*; *Walmsley et al., 2013a*; *Walmsley et al., 2013b*). The goal of this study is to apply both FEA and methods of quantifying long bone cross-sectional geometry to the forelimb of Australian mekosuchine crocodiles in order to help resolve their locomotion in a quantitative context.

The endemic radiation of Australasian mekosuchines, which currently consists of ten genera reported from sites spanning the Eocene through Plio–Pleistocene, produced a variety of cranial shapes during a period of morphological diversification in the Oligo–Miocene reflecting vigorous niche specialisation (*Willis, 1993*; *Willis, 1997a*; *Megirian, 1994*). Among these specialisations, a greater emphasis on terrestrial hunting not seen in extant crocodylian communities has been suggested (*Willis, 1993*; *Willis, 1997a*; *Megirian, 1994*). Evidence to support this consists of cranial features such as labiolingually compressed 'ziphodont' teeth and reorientation of the orbits and nares anteriad on the cranium (*Willis, 1993*; *Willis, 1997a*; *Megirian, 1994*). If such terrestrial specialisation were the case, it can be predicted that similar specialisations for locomotion might develop in the postcranial skeleton, particularly with regards to sprawling versus high-walk gaits. Extant crocodylians employ these gaits variably to suit different ecological needs (*Reilly & Elias, 1998*). Sprawling gait, where the limbs are abducted from the body, is adopted when moving between land and water (*Reilly & Elias, 1998*). High-walk gait, where the limbs are held nearly parasagittal under the body, is adopted for prolonged movement overland (*Reilly & Elias, 1998*). Previously reported features of the limbs and pelves do suggest that locomotory changes were occurring in mekosuchine postcrania, at least from
the standpoint of the biomechanics of the orientation of the musculature (*Stein et al., 2012*). The earliest mekosuchine humeri from Eocene sediments at Murgon, southeastern Queensland, differ morphologically from those of extant crocodylians, displaying very little offset of the proximal extremity and deltopectoral crest against the diaphyseal shaft, potentially increasing the out-lever force of limb adductors such as the pectoralis (*Stein et al., 2012*).

Curvature in the long-bones of tetrapod limbs presents a paradox. The ideal structure to withstand strain is essentially columnar, yet some degree of curvature is ubiquitous in the diaphyses of tetrapods (*Lanyon, 1980*; *Meers, 2002*). Curvature in individuals can partially be attributed to the effect of remodelling bone in response to muscle action through life, an effect observable in limbs subjected to habitual motion (*Lanyon, 1980*; *Bass et al., 2002*; *Taylor et al., 2009*). The drivers of variation over evolutionary timeframes between tetrapod genera, however, are less clear. Curvature, and proportional variation in general, appears independent of the scaling of bodyweight to bone volume across the long bones of tetrapods (*Campione & Evans, 2012*). Several different, although not necessarily exclusive, hypotheses have been proposed to explain interspecies variation in curvature. Experimental evidence indicates that curvature sacrifices absolute resistance to strain while confining functional strains to a single axis, increasing the predictability of failures and maximising metabolic efficiency by directing reinforcement through bone remodelling over an individual's lifetime (*Lanyon, 1980*). Curvature in long bones has also been shown to act as a pre-buckled strut in opposition to the unavoidable internal strains imposed by the action of limb musculature for locomotory strategies (*Bertram & Biewener, 1988*; *Milne, 2016*). Variation in curvature also might relate to factors other than strain, such as allowing physical space for muscle belly packing (*Cubo, Menten & Casinos, 1999*).

Mekosuchine limbs can be interpreted within this theoretical framework. Muscle packing is a potential factor, but often beyond the scope of the fossil record to confirm. Changes in muscle-body volume are a possible outcome of the altered lines of action for the limb musculature, due to changes in the orientation and positions of muscle origins and insertions, but would require evidence of ligamentation to confirm. A divergence in gross morphology cannot be ruled out, although the musculature of crocodylians appears to be generally conservative, even over long timeframes (*Meers, 2003*). Bone shape characteristics and their relation to functional strains, however, can be readily assessed. There is evidence that the crocodylian sprawl is secondarily derived with the adoption of semi-aquatic habitus (*Parrish, 1986*; *Parrish, 1987*; *Sereno, 1991*; *Reilly & Elias, 1998*). As a result of this adoption, the long bones of extant crocodylian limbs (i.e., humerus and femur) are loaded primarily under torsional bending, unlike the parasagittal limbs of mammals and birds which are loaded primarily under axial bending (*Blob & Biewener, 1999*; *Blob & Biewener, 2001*; *Blob et al., 2014*). It can be hypothesised that the pronounced offset between extremities, as exemplified in the sprawling gait of extant eusuchians, is an adaptation for strain mitigation in a limb subject to torsional strains, as per the predictability and pre-buckled strut theories noted above. The trade-off under this scenario, however, appears to be increased axial bending moments and hence stress during the high-walk when the limbs are held closer to the sagittal plane, as has been observed in the alligator

hind-limb (*Blob & Biewener, 1999*; *Reilly & Blob, 2003*). Do mekosuchine limbs display a difference in shape and response to locomotory loads (i.e., structural stresses) that might indicate locomotory emphasis other than semi-aquatic habit?

There have been few previous studies examining how limb morphology relates to trophic specialisation in crocodylians. Limb length has been found to correlate with trophic habits in extant crocodylians, with more piscivorous genera displaying shorter limbs (*Iijima, Kubo & Kobayashi, 2018*). Analysis of cross-sectional measurements of the diaphysis of the crocodylian humerus did not show a significant difference in cross-sectional proportion between extant crocodiles and alligators, using traditional geometric measures of proportion and resistance to bending, including $Ix/Iy$, $Imax/Imin$ and $J$ (*Meers, 2002*). $Ix/Iy$ measures the second moment of area about the $X$-axis relative to the $Y$-axis and $Imax/Imin$ measures the first and second principal moments of area about the centroid and principal axis. Together these measures approximate rigidity along dorsoventral, craniocaudal and mediolateral planes, the latter accounting for the planes of maximum resistance to bending. $J$ is a measure of torsional rigidity and is usually considered the most important cross-sectional measure of strength from a biological perspective (e.g., *Shaw & Stock, 2009*). Taken together, these biomechanical property measures provide a way to discern shape difference quantitatively and are strongly linked to behaviour across a wide spectrum of taxa (e.g., *Hayashi et al., 2013*; *Straehl et al., 2013*; *Amson et al., 2014*). These relationships reflect bone's response to loading conditions through changes to dimensions of the cortex (*Lanyon & Rubin, 1984*; *Frost, 2003*) and have been used as a basis for inferring locomotory capacity for extinct taxa, providing information for palaeobiological reconstructions (e.g., *Botha & Chinsamy, 2004*; *Kolb et al., 2015*; *Klein et al., 2016*).

Humeri are the best-preserved elements of the mekosuchine limb currently available for testing. Therefore, in this study, humeri of the extant semi-aquatic saltwater crocodile *Crocodylus porosus* (*Schneider, 1801*), freshwater crocodile *Crocodylus johnstoni* (*Krefft, 1873*), and those available for mekosuchines from Eocene to Pleistocene Australian fossil sites were compared using finite element analysis and geometric morphometric methods to analyse the loading consequences of cross-sectional shape variance in the diaphysis (*Wilson & Humphrey, 2015*). The latter method accounts for entire cross-sectional shape variation and is combined here with biomechanical property data to assess the biomechanical implications of differences between taxa in the amount and distribution of cortical bone. For the finite element analysis, specimens were tested under two loading scenarios, representative of sprawling and high-walking gaits. If mekosuchine humeri were subject to similar locomotor regimes and therefore the same imperatives for stress mitigation as those of extant semi-aquatic crocodylians, it can be predicted that there will be no difference in diaphyseal cross-sectional shape or patterns of stress under sprawl and high-walk. A rounded cross-sectional shape might be expected in this case, as has been observed in American alligators and theorised to be highly resistant to torsional loads that predominate in extant crocodylians (*Blob & Biewener, 1999*; *Meers, 2002*).

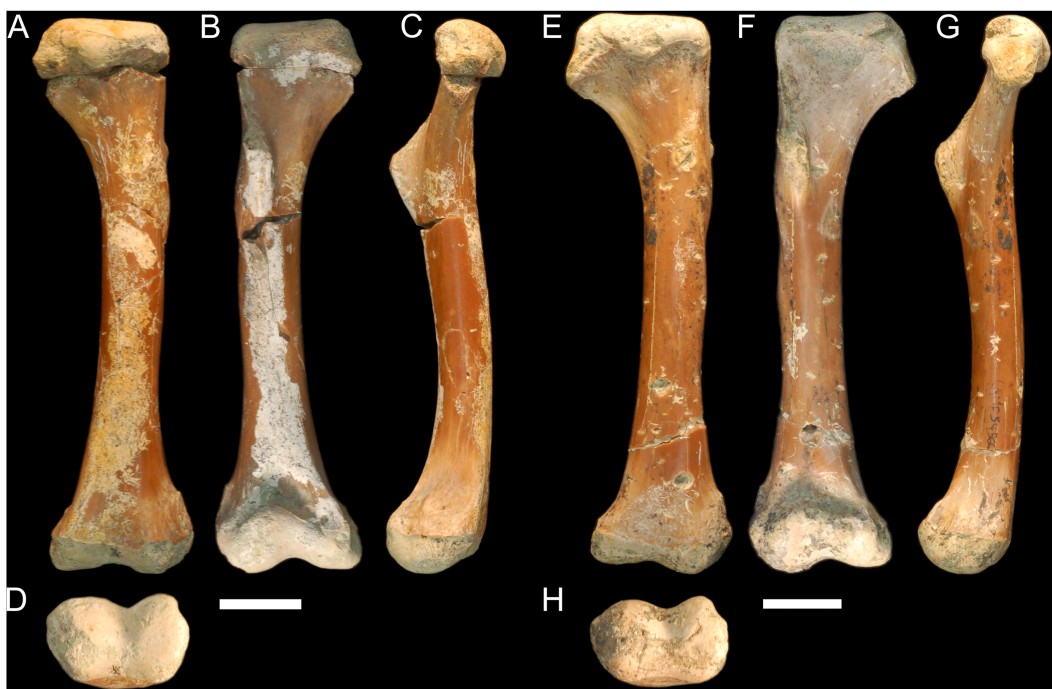

**Figure 1** **Fossil mekosuchine crocodile specimens from the Australian Eocene Murgon locality used in study.** Right humerus QM F56058 in (A), dorsal (B), ventral (C), medial and (D), distal views. QM F56060 in (E), dorsal (F), ventral (G), medial and (H), distal views. Scale bars = 2 cm. Specimens were originally published in *Stein et al. (2012)* and are reproduced here with permission.

## METHODS & MATERIALS

### Examined Materials

Seven mekosuchine humeri; QMF56058, QM F56060, NTM P907-70, QM F57954, QM F57955, QM F57956, and QM F57953 were available and sufficiently complete for use in this study. Three humeri of sub-adult to adult extant saltwater and freshwater crocodiles XCb Cp4, AR 22025 and AR 22161 were also included in this study.

QM F56058 and QM F56060 (Figs. 1A–1H). Left and right humeri referable to the genus *Kambara* from Murgon, southeastern Queensland (*Salisbury & Willis, 1996*). The Tingamarra Local Fauna at Murgon is early Eocene in age, radiometrically dated as 54–55 Ma (*Godthelp et al., 1992*). These specimens were first described in (*Stein et al., 2012*). The diaphysis is elongate and distinctly columnar in aspect, displaying minimal curvature along its length. The proximal and distal extremities align parallel to one another in the dorsoventral plane. The deltopectoral crest extends from the lateral edge of the diaphysis perpendicular to the proximal extremity and forms a wide triangle. The medial and lateral epicondyles of the distal extremity are equal in size, with the boarder of the lateral condyle situated nearly above condylar sulcus.

NTM P907-70 (Figs. 2A–2D). Right humerus collected from Top Site, Bullock Creek, Northern Territory, in association with materials referable to the genus *Baru* (*Willis, Murray & Megirian, 1990*). Stratigraphy and bio-correlation estimate the Camfield beds to

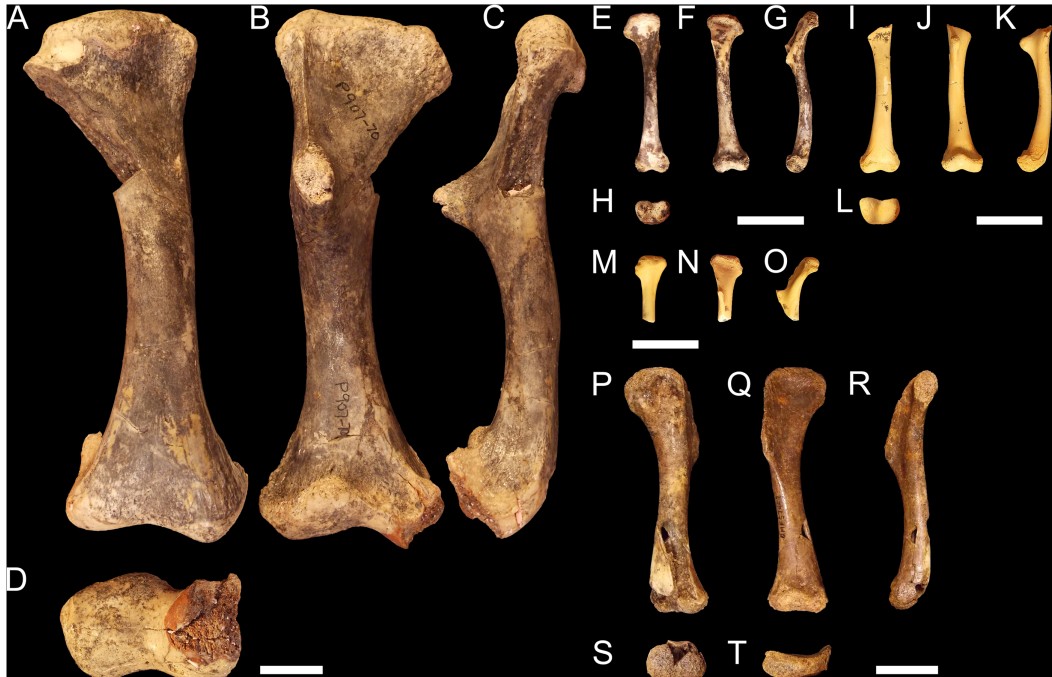

**Figure 2** **Fossil mekosuchine crocodile specimens from the Australian Oligo—Miocene Riversleigh, Bullock Creek and Plio—Pleistocene Floraville localities used in study.** Right humerus NTM P907-70 in (A), dorsal (B), ventral (C), medial and (D), distal views. QM F57954 in (E), dorsal (F), ventral (G), medial and (H), distal views. QM F57955 in (I), dorsal (J), ventral (K), medial and (L), distal views. QM F57956 in (M), dorsal (N), ventral and (O), medial views. QM F57953 in (P), dorsal (Q), ventral (R), medial (S), distal and (T), proximal views. Scale bars = 2 cm.

be middle Miocene in age, approximately 12 Ma (*Murray et al., 2000*; *Black et al., 2012*). The humerus is robustly proportioned, with mediolaterally wide proximal and distal extremities. The extremities and diaphysis display the same distinctly columnar aspect as the humeri associated with *Kambara* species, in that the extremities align with one another in the dorsoventral plane. NTM P907-70 is distinguished by a slight medial offset of the proximal extremity from the distal extremity, as well as a more distinct dorsoventral curvature along the ventral diaphysis. The deltopectoral crest is dorsoventrally tall and extends perpendicular from the diaphyseal shaft, but is very compact in proximodistal length, forming an acute triangle. Medially opposite the crest, the diaphysis displays a distinct thickening. The medial and lateral epicondyles of the distal extremity distinctly elongate dorsoventrally and there is a pronounced asymmetry in the size of the medial and lateral condyles, the medial condyle being larger of the two and flattening proximodistally into a broad surface. The border of the lateral condyle is situated nearly above the condylar sulcus.

QM F57954, QM F57955 and QM F57956 (Figs. 2E–2O). Three humeri collected from Ringtail Site, Gag Plateau, Riversleigh World Heritage Area, northwestern Queensland (*Archer et al., 2006*). Ringtail Site has been radiometrically dated as 14–13 Ma or middle Miocene in age (*Woodhead et al., 2016*). Three mekosuchines have been reported from

Ringtail Site, *Trilophosuchus rackhami* (*Willis, 1993*), *Mekosuchus sanderi* (*Willis, 1997a*) and indeterminate remains of *Baru* species (*Willis, 1997a*). All three humeri are relatively small, less than 5 cm in proximodistal length. In contrast to their size, the deltopectoral crest is well developed, similar in shape to the acute crest displayed by *Baru* species in NTM P907-70 (see above) and extending perpendicular from the diaphysis. The attachment for the m. pectoralis is particularly well developed and rugose. As for *Kambara* and *Baru* species, the proximal and distal extremities align in the dorsoventral plane and the diaphysis displays a similar profile as NTM P907-70. The distal extremity does not display pronounced asymmetry in size between the condyles and the border of the lateral condyle is situated directly over the condylar sulcus. The similar proportions of the deltopectoral crest would suggest that these humeri represent juvenile individuals. The specimens are somewhat differently proportioned, being more gracile than the highly robust NTM P907-70. Allometry during ontogeny is a plausible explanation for the difference in proportion. It is also possible that these humeri may represent other Oligocene Riversleigh mekosuchines (*Trilophosuchus*, *Mekosuchus* or *Ultrastenos*).

QM F57953 (Figs. 2P–2T). Humerus collected from Henk's Heap Site, Floraville, northwestern Queensland. Stratigraphy and bio-correlation suggest these sediments are early Pleistocene in age, approximately 1–2 Ma (*Grimes & Doutch, 1979*; *Rich et al., 1991*; *Klinkhamer & Godthelp, 2015*). The only mekosuchine genus reported from the Australian mainland in this period is *Pallimnarchus*. The humerus is small, about 8 cm in dorsoposterior width, and thus may represent a young adult individual of a *Pallimnarchus* species. The deltopectoral crest is strongly developed in proximodistal length forming a broad wedge that runs from the proximal head to nearly half the length of the humerus. The deltopectoral crest extends 100-—120° from the line of the proximal extremity, well beyond even the perpendicular orientation displayed by pre-Pleistocene mekosuchine humeri. In addition, the proximal extremity is displaced medially to accommodate the large size of the deltopectoral crest. As a result, the proximal extremity is a broad, open, fanned structure. Despite this, the proximal and distal extremities align in the same dorsoventral plane relative to one another as other mekosuchine humeri and the diaphysis displays minimal curvature. The epicondyles of the distal extremity are indistinct, possibly due to erosion, but appear equally sized and the boarder of the lateral condyle is situated above condylar sulcus.

The large time-frame under consideration raises the possibility of convergence in locomotory habit among mekosuchine communities. Because any potential homoplasy within the subfamily Mekosuchinae is of interest to this study, analyses did not correct for phylogenetic relatedness. As no other modern lineage aside from Plio–Pleistocene *Crocodylus* has been reported from Tertiary Australian localities (*Willis, 1997b*), the impact on the results of homoplasy outside the subfamily should be minimised.

## Analysis
### CT scanning and image processing
Micro computed tomography (CT) scans were obtained for specimens using a Siemens Inveon Multimodality Platform micro CT scanner and a Siemens Syngo CT2012B in

**Table 1  CT scan protocols used in study.** Scans of specimen XCb Cp4 were performed using a Siemens Syngo Definition AS+ scanner using photographic interpretation set to monochrome 2, and CW rotation direction. Scans of all other specimens were performed using a Siemens Inveon Multimodularity (MM) Platform scanner using the Feldcamp Cone Be algorithm, scanner options set to Acq Mode 9, photographic interpretation set to monochrome 2, and CC rotation direction.

| Specimen | ExT | Px Size (mm) | SlI (mm) | SlT (mm) | #Slice | FoV (mm) | Width (px) | Height (px) | Dist StD (mm) | Dist StO (mm) |
|---|---|---|---|---|---|---|---|---|---|---|
| XCb Cp4 | 1000 | 0.820312 | 0.750 | 0.750 | 3019 | 420.00 | 512 | 512 | 1085.6 | 595 |
| AR 22025 | 300 | 0.071925 | 0.072 | 0.072 | 1529 | 55.24 | 768 | 768 | 3459 | 1838 |
| AR 22161 | 300 | 0.071925 | 0.072 | 0.072 | 2085 | 55.24 | 768 | 768 | 3459 | 1838 |
| QM F56058 | 300 | 0.071925 | 0.072 | 0.072 | 2085 | 55.24 | 768 | 768 | 3459 | 1838 |
| QM F56060]F56060 | 300 | 0.071925 | 0.072 | 0.072 | 2085 | 55.24 | 768 | 768 | 3459 | 1838 |
| QM F57954 | 2200 | 0.027735 | 0.028 | 0.028 | 2162 | 28.4 | 1024 | 1024 | 3509 | 1438 |
| QM F57955 | 2200 | 0.027735 | 0.028 | 0.028 | 1694 | 28.4 | 1024 | 1024 | 3509 | 1438 |
| NTM P907-70 | 250 | 0.106305 | 0.106 | 0.106 | 1565 | 67.61 | 636 | 636 | 3359 | 2638 |
| QM F57953 | 290 | 0.05547 | 0.055 | 0.055 | 1532 | 28.4 | 512 | 512 | 3509 | 1438 |

**Notes.**

ExT, exposure time; PX Size, pixel size; SlI, slice increment; SlT, slice thickness; #Slice, number of slices; FoV, field of view; Dist StD, distance source to detector; Dist StO, distance source to object.

the case of XCb Cp4 (see *Klinkhamer et al., 2017*) for details). Protocols for each scan are detailed in Table 1. DICOM slices were converted into digital 3D isosurface models using the Mimics v.18 and 3-Matic v.16 software packages (*Materialise, 2015a*; *Materialise, 2015b*). Bone voxels of each slice were isolated using the mask function at high quality setting in Mimics. Masks were manually checked and edited where necessary to ensure accuracy and remove unavoidable artefacts of the CT scanning process. Measurements to the nearest millimetre were compared between the resulting 3D isosurface models and their corresponding original specimen to verify the 3D model was an accurate representation of the original geometry. 3D isosurfaces were then used to generate solid mesh models using 3-Matic. Solid meshes were generated at a final resolution of around one million tetrahedral elements, following protocols established for comparative FEA studies (*McHenry et al., 2007*; *Attard et al., 2014*; *Attard et al., 2016*). In the case of the Riversleigh humeri, a single whole solid mesh was generated in Mimics by matching the complete proximal extremity of QM F57954 to the incomplete base of QM F57955, and confirming the alignment with QM F57956. In the case of NTM P907-70 and QM F57953, damaged portions were reconstructed by warping a mesh of *Kambara* specimen QM F56060 using the Landmark IDAV v3.0 software package (*Wiley et al., 2005*) to curve and patch elements placed at homologous landmarks, detailed in Article S1, following the method of *Parr et al. (2012)*.

## Extraction of virtual cross sections from 3D models

Virtual cross sections were extracted from the 3D isosurface models using Rhinoceros v.5 software (*Robert McNeel & Associates, 2012*) following the protocol outlined by *Wilson & Humphrey (2015)*. To summarize, 3D isosurface models were initially oriented in virtual space to ensure an absolute frame of reference and to eliminate variation in pitch, roll and translation between models. The principle axis of inertia of each model was aligned and fixed to the digital $z$-axis. The axis running between lateral and medial condylar apexes of the distal extremity was aligned to the digital $x$-axis perpendicular to the $z$-axis. As

such, the digital $x$, $y$ and $z$ axes represent the craniocaudal, dorsoventral and lateromedial anatomical frames of reference for the humeri respectively. The centroid of each model was set to (0,0,0) in the universal coordinate system. Oriented models were then virtually sectioned. The bounding box of each model was calculated, and a virtual cross section was extracted from the 3D model at 50% of the total bounding box length (i.e., midshaft), parallel with the digital $x$-axis. The centroids of each section were aligned to (0,0,0) in the universal coordinate system and a series of 16 equiangular radii were projected from each centroid, with the first radii aligned to (0,1,0). Cartesian semi-landmarks were generated where radii intercepted with the cross section at the periosteal margin, resulting in a set of spatially-homologous landmark data for each specimen represented by 16 periosteal semi-landmarks. Landmark data for each specimen were exported from Rhinoceros 5 as point files for further analyses (*Wilson & Humphrey, 2015*).

## Analysis of cross-sectional shape and biomechanical properties

Landmark data sets were subject to Procrustes superimposition to remove the effects of scaling, rotation and size between configurations (*Rohlf & Slice, 1990*). Interspecific differences in allometric growth patterns in cross-sectional shape have also been reported among extant crocodylians (*Dodson, 1975*; *Meers, 2002*; *Iijima & Kubo, 2019*). The estimation of allometric growth equations, requiring multiple individuals of different ontogenetic stages, is not possible for the available mekosuchines herein. However, to account for allometric scaling, due to the considerable difference in size between specimens (Figs. 1 and 2), residuals were extracted from a multivariate regression of Procrustes coordinates on log body mass (shape~size). These regressions were conducted in R v. 3.6 using the package 'geomorph' v. 3.1.3 and the function *procD.lm* (*Adams & Otárola-Castillo, 2013*). Body mass was estimated for each specimen (Table 2), using the humeral circumference regressions of *Campione & Evans (2012)*. Differences in periosteal shape were compared between extant and fossil crocodylians using ordinations from principal component analysis (PCA). The broken stick model (*Jackson, 1993*) was used to determine the number of principal components (PCs) explaining significant portions of variance in the sample.

In addition to the landmark data, cross-sectional biomechanical property data were extracted using the Moment Macro Plugin v.1.2 for ImageJ version 1.51 g (*Rasband, 2011*; *Ruff, 2016*), following *Wilson & Humphrey (2015)*. These comprised the traditional measures of bending strength that have been applied in studies of bone function adaptation (e.g., *Trinkaus, Churchill & Ruff, 1994*; *Ruff, Holt & Trinkaus, 2006*; *Shaw & Stock, 2009*). Measured values were: $Ix/Iy$, $Imax/Imin$ and $J$, cortical area and theta, the angular measurement of orientation of greatest bending rigidity. Values of $Ix/Iy$ were computed to measure bending strength in the anterior-posterior plane relative to the medial-lateral plane. A value of 1 for $Ix/Iy$ would indicate that the cross section is circular, whereas values of <1 indicate greater bending strength along the medial-lateral plane (medial-lateral diameter >anterior-posterior diameter) and values of >1 reflect greater bending strength along the anterior-posterior plane (anterior-posterior diameter >medial-lateral diameter). Values of $Imax/Imin$ are section-orientation invariant and are always equal to or greater

**Table 2  Specimen body mass estimates.** Body mass estimates deduced for each sample in kg following the humeral circumference regressions of *Campione & Evans (2012)*.

| Specimen | Taxon | Period | Locality | Estimated Body Mass (kg) |
|---|---|---|---|---|
| XCb Cp4 | *C. porosus* | Recent | Queensland | 129.94 |
| AR 22025 | *C. johnstoni* | Recent | Queensland | 67.22 |
| AR 22161 | *C. johnstoni* | Recent | Queensland | 93.10 |
| QM 56058 | *Kambara* sp. | Eocene | Murgon | 83.80 |
| QM 56060 | *Kambara* sp. | Eocene | Murgon | 85.71 |
| NTM P907-70 | *Baru* sp. | Miocene | Bullock Creek | 130.63 |
| QM F57954 | ?*Mekosuchus* sp./ ? *Baru* sp. | Miocene | Riversleigh | 16.46 |
| QM F57955 | ?*Mekosuchus* sp./ ? *Baru* sp | Miocene | Riversleigh | 19.29 |
| QM F57953 | ?*Pallimnarchus* sp. | Pleistocene | Floraville | 47.45 |

than 1.0 and, similar to $Ix/Iy$, measure the relative bending strength along two planes. Values of closer to 1.0 for Imax/Imin values indicate circular cross-section, with the minimum and maximum planes of bending resistance being relatively similar in magnitude to each other. Larger values indicate elliptical cross-section, with the maximum plane increasing in magnitude over the minimum plane (*Ruff, 1987*). *J* measures the polar moment of area about the medio-lateral axis, with larger values indicating greater resistance to torsional loads. Theta is defined as the counter clockwise angle from the medial-lateral axis to the *Imax* axis (*Ruff & Hayes, 1983*). These values were compared using reduced major axis (RMA) regression, selecting variables following the regressions reported by *Meers (2002)*. RMA regression was conducted in R v. 3.6 using the package 'smatr' v. 3.4.-8 and the function sma (*Warton et al., 2012*). As landmark data were unavailable for QM F57956, this specimen was excluded from the shape analysis. QM F57955 was excluded from analyses requiring full humeral length. Due to CT artefacts from the bonding agent, the endosteal shape for NTMP907-70 had to be manually traced, and this prevented calculation of theta due to software limitations.

## Finite element modelling

Solid meshes were imported into the Strand7 (*Strand7 PtyLtd, 2013*) finite element analysis package as Nastran (.nas) files. Meshes were assigned material properties for cortical bone, with a Young's modulus of 13.7 GPa and Poisson's ratio of 0.4 (*McHenry et al., 2007*; *Wroe et al., 2007*; *Wroe, 2008*; *Wroe et al., 2008*; *Wroe, 2010*). However, it is important to note that in a wholly comparative context, as used here, the actual modulus is unimportant (*Wroe et al., 2018*). Two loading regimes were used to simulate forces of the stance phase of locomotion in crocodylians during sprawling and high-walking gaits. Kinematics appear nearly identical between sprawl and high-walk, the main difference being the relative level of abduction of the limb (*Reilly & Elias, 1998*). The first loading case modelled the humeri during the sprawl, placing the long axis +8° above the transverse plane. The second loading case modelled the humeri during the high-walk placing the long axis −68° below the transverse plane. These values represent the maximum and minimum humeral

abduction angles established by previous observations of kinematics in crocodylians (*Reilly & Elias, 1998*; *Hutchinson & Gatesy, 2000*; *Baier & Gatesy, 2013*).

## Boundary conditions

$X$, $Y$ and $Z$ coordinate axes were maintained from the bone positioning required for the extraction of virtual cross-sections, such that the humeri retained identical orientation in Strand 7. Humeri were fixed proximally and distally to simulate cartilaginous articulation with the scapulocoroacoid and antebrachium. Proximally the humerus was fixed along the ventral surface of the proximal extremity at a row of nodes simulating the contact with the glenohumeral facet of the scapulocoroacoid. Distally the humerus was fixed at three nodes to simulate contact with the radius and ulna. The first node was placed centrally in the sulcus between condyles where the cartilaginous intercotylar crest and process articulates dorsally with the sulcus, accounting for the action of the crest at maximum extension of peak stance phase to prevent lateral movement of the humerus. The second and third nodes flank the first on the ventral surfaces of the lateral and medial condyles. All nodes were fixed against translation and rotation in all three axes. To disperse the generated point forces, the fixed nodes were surrounded by a lattice of beams given the properties of structural steelwork and geometric diameter of 2.5 mm, following the method of *Attard et al. (2016)*. Restraining lattices were restricted to the same proportional surface area to prevent over restraining the model, usually accounting for 1,000–2,000 beams in larger models and 100–200 beams in smaller models.

## Loading conditions

Muscles were simulated as trusses attaching to the humerus via hexagonal lattices comprised of twelve beam elements to disperse forces along the mesh (*Attard et al., 2016*). Attachment lattices were given the properties of structural steel work and geometric diameter of one millimetre. Muscle trusses were assigned a Youngs modulus of 1 MPa (*McHenry et al., 2007*; *Wroe et al., 2007*; *Wroe, 2008*; *Wroe et al., 2008*; *Wroe, 2010*), and geometric diameter of 0.25 millimetres. Muscle trusses were orientated and anchored away from the humerus to match with their orientation during the stance phase of the step cycle, and were then pre-tensioned with forces that simulated the contraction of the stance phase musculature. Initially muscle force estimates for the freshwater crocodile specimen AR14934 were calculated from previously recorded physiological cross-sectional area values for each muscle (*Porro et al., 2011*; *Allen, Molnar & Parker, 2014*). To account for allometry and the effects of body size, these muscle forces were then scaled to the estimated body masses for the saltwater crocodile and mekosuchine specimens under a 2/3 power law using the body mass of AR14934 as reference (*McHenry et al., 2007*; *Wroe et al., 2007*; *Wroe, 2008*; *Wroe et al., 2008*; *Wroe, 2010*; *Klinkhamer et al., 2019*).

Electromyographic studies of the hindlimb in alligators show that functional muscle groups contract near simultaneously at the peak of stance phase (*Gatesy, 1997*; *Reilly et al., 2005*). Assuming a similar situation in the humerus, muscle action during the stance phase was modelled as three functional groups acting simultaneously on the models. The first functional group consisted of those muscles that act on the proximal head to flex the

humerus, providing propulsion during the stance phase (*Meers, 2003*). This functional group can be subdivided into those muscles that act alternately, ventrally and dorsally on the humerus. The ventral group are the primary actors for retracting the humerus, consisting of the paired actions of the m. pectoralis and m. coracobrachialis brevis ventralis on the ventromedial surface of the deltopectoral crest. Dorsally the m. teres major elevates the humerus and exerts an effectively caudal strain across the proximal head.

The second functional group consists of the action of the m. triceps brachii group. This complex group acts to extend the antebrachium during the stance phase (*Meers, 2003*). Composed of five muscles in crocodylians, the triceps brachii originate alternately from the scapulocoracoid or the diaphysis of the humerus and insert onto the ulna by a shared tendon that loops over the humerus/ulna joint. Because of this pulley-like insertion, the action of the triceps brachii effectively exerts compressive stresses both longitudinally along the diaphysis of the humerus and ventrally onto its distal extremity.

The third functional group consists of those muscles that act on the distal extremities in concert with the m. triceps brachii to extend and supinate the manus during the stance phase (*Meers, 2003*). These include the m. flexor ulnaris, m. extensor carpi radialis longus, the m. extensor carpi ulnaris longus and the m. supinator. When these muscles act, they apply a ventrally directed stress to the lateroventral surface of the lateral condyle in addition to the stress generally applied by the triceps brachii.

One further loading accounted for is the external load applied to the limb by the weight of the body. This was modelled as a ventrally directed load acting on a lattice of beams given the properties of structural steelwork and geometric diameter of 2.5 at the proximal end of the humerus, with the nodes fixed against translation and rotation in the X and Z axes so that the force was only applied in a ventral direction. As for the restraining lattices, the number of beams used was proportional to the size of the model, approximately 300 in larger models and 50 in the smaller models. Newton values of this load were calculated from estimated body mass. To separate out the effects of internal musculature load versus the external load of body mass two analyses were run consisting of the loadings for limb musculature alone and limb musculature plus body mass for all three loading regimes.

To provide a comprehensive view of any variation in stress patterns between models, both Von Mises (VM) stresses and principle compressive and tensile stresses were recorded as colour-contour maps for each model and by taking samples using the peek tool in Strand 7. VM stress values were extracted at (1) the proximal and (2) distal extents of the deltopectoral crest, (3) lateral and (4) medal extents of the dorsal surface of diaphysis in line with the apex of the deltopectoral crest, (5) dorsally and (6) ventrally on the mid diaphyseal shaft and (7) dorsally on the lateral and (8) medial epicondyles. These locations were chosen so that they did not overlap with those regions of the model that were constrained or had force directly applied, to avoid artefacts. At each point, six values were extracted from bricks and averaged, these comprised the chosen point and five adjacent, surrounding bricks.

## RESULTS

Available specimens of fossil mekosuchine humeri indicate that the columnar aspect observed in Eocene *Kambara* taxa persisted through the Cenozoic. Additionally, novel morphologies appear to have developed, with the humerus of at least one mekosuchine taxon coming to resemble those of Mesozoic notosuchids and neosuchids (*Pol, 2005*; *Sereno & Larsson, 2009*; *Sertich & Groenke, 2010*; *Riff & Kellner, 2011*; *Pol et al., 2012*). Measures of $Ix/Iy$, $J$, cortical area, and humeral length extracted from mekosuchine specimens are plotted in Fig. 3 following *Meers (2002)*. When compared, these appear to describe a scaling relationship between cortical area and resistance to bending in line with beam theory and what has been previously been reported in crocodylians, returning RMA regression gradients that are similar to those of *Meers (2002)* (compare Figs. 6 [4.723], 7 [1.022], 8 [0.505] and 3 [2.361] of that publication to Figs. 3A [3.659], 3B [0.961], 3C [1.364], 3D [2.651] here). $R^2$ and $P$-values along with confidence intervals calculated for the RMA regressions are provided in Table 3. $J$ positively correlates with cortical area and although beam theory would predict that $J$ should decrease with humeral length, the proportionally large cortical areas observed in crocodylian humeri appear to counteract this (*Hildebrand & Goslow, 1998*; *Blob & Biewener, 1999*). The notable difference from the gradients reported by *Meers (2002)* is a greater increase in $J$ per increase in cortical area, but a slightly lower increase in $J$ per increase in humeral length (Figs. 3A and 3C). The confidence intervals for RMA regressions of humeral length are large, however, and span the value reported by *Meers (2002)*.

Results of the principal component analysis are plotted in Fig. 4 along with illustrations of extracted periosteal shapes at extreme positive and negative values along the PC axes. PCA eigenvalues, scores and loadings are reported in Article S1. The saltwater and freshwater crocodile specimens (in red) group centrally on PC1 (69.1%, Figs. 4A–4B) which describes the orientation of an axis of lateral compression of the diaphyseal shaft, but extend into negative values along PC3 (7.09%, Fig. 4B) which describes circular versus square periosteal shape, reflecting a comparatively rounded periosteal outline. Most of the mekosuchine specimens (in blue) diverge along PC1 and group with positive values on PC3 with the exception of QM F57954, reflecting some degree of flattening of periosteal outline, resulting in an elliptical shape. PC2 (20.7%, Fig. 4A), which describes the extent of axial flattening, displays considerable variation across all specimens. This is corroborated by the wide spread of $Imax/Imin$ values extracted from the periosteal outlines, plotted in Fig. 5, which can be expected to be particularly sensitive to this variation of shape (Fig. 5A), and the variation in theta between specimens (21.29°–−50.52°, range = 71.81°) where values closer to 0° indicate a rounded outline (Fig. 5C).

Results of the finite element analyses are visualised in Figs. 6–8 and average extracted stresses are plotted in Figs. 9–10. Stresses tend to be confined to the proximal extent of the diaphysis in the mekosuchine models. In contrast, stresses are distributed though the diaphysis in a sinusoidal pattern in the saltwater and freshwater crocodile models. Generally, the shift between sprawling and high-walk gaits results in an increase in structural stresses, consistent with what has been observed in alligators by *Blob & Biewener (1999)*; *Blob &*

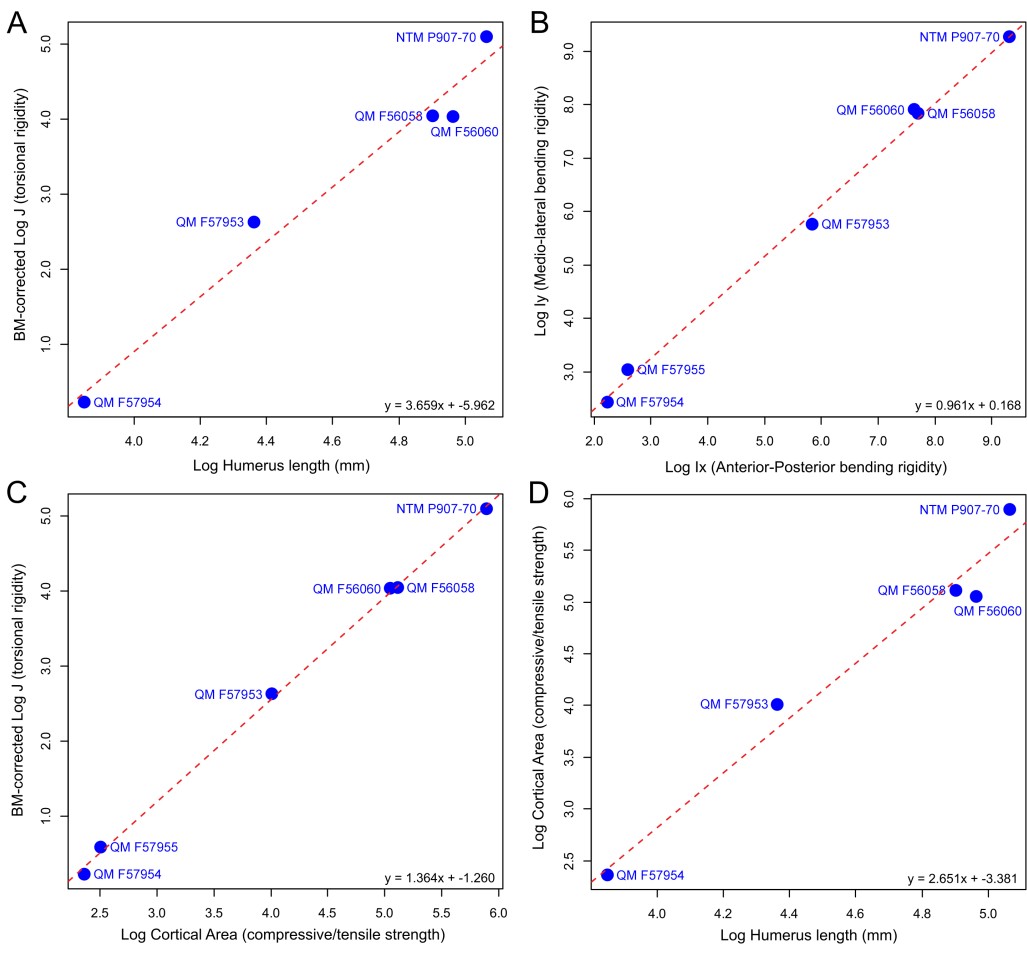

**Figure 3** **Plots of biomechanical property measurements extracted from diaphyseal cross sections of 3D models of the humerus at 50% (midshaft).** Measures, after *Meers (2002)*, are (A) torsional rigidity versus humeral length, (B) mediolateral rigidity versus anteroposterior rigidity, (C) torsional rigidity versus cortical area, (D) cortical area versus humeral length. These measures provide information on the overall rigidity of the humerus and cross-sectional distribution patterns for cortical bone (mediolateral versus anteroposterior plane) in relation to bone length.

*Biewener (2001)* (Figs. 6, 7, 8 and 9). Von Mises (VM) stresses of the mekosuchine humeral models are generally lower than the those of the extant crocodile models and the increase in VM stresses between sprawl to high-walk is generally much smaller, for example a 2% (approximately 0.2 MPa) increase in combined VM stresses in the Murgon QM F56060 model compared to 20–25% (approximately 5–10 MPa) increases in the saltwater and freshwater crocodiles (Figs. 9A–9C). Peak stresses for the saltwater and freshwater crocodile models are within the range of *in-vivo* stresses measured in alligator femurs (*Blob & Biewener, 2001*), suggesting the chosen protocols have resulted in an accurate estimate of structural stresses.

**Table 3   Results of reduced major axis regressions for cross-sectional geometry properties of the humerus for mekosuchine taxa.** Regression models A–D are graphed in Figs. 3A–D

| Regression model (y ∼x) | N | $R^2$ | *P*-value | Slope (Confidence Interval) | Intercept (Confidence Interval) |
|---|---|---|---|---|---|
| A. J ∼Humerus Length | 5 | 0.97 | 0.0024 | 3.66 (2.65–5.05) | −5.96 (−8.39–−3.53) |
| B. Ix ∼Iy | 6 | 0.99 | <0.0001 | 0.96 (0.88–1.05) | 0.17 (−0.06–0.39) |
| C. J ∼Cortical Area | 6 | 0.99 | <0.0001 | 1.36 (1.30–1.43) | −1.26 (−1.38–−1.14) |
| D. Cortical Area ∼Humerus Length | 5 | 0.97 | 0.0024 | 2.65 (1.92–3.66) | −3.38 (−5.13–−1.63) |

# DISCUSSION

The results of the geometric morphometric analyses indicate differences in cross-sectional shape of the diaphyseal midshaft exist between mekosuchines and extant crocodiles. As predicted, the saltwater and freshwater crocodile specimens group towards rounded shapes in the principal component analysis, conforming with what has been observed previously in alligator limb bones (*Blob & Biewener, 1999*; *Meers, 2002*). In contrast, the majority of mekosuchine specimens appear to group into a different area of morphospace along PC1 and PC3 that describes a series of elliptical cross-sections. These elliptical cross-sections can be related to shifts in the plane of maximum resistance to bending. The mekosuchine specimens appear to form a trend when PC1 is compared with *Ix/Iy* (Fig. 5B) and display a wide range of theta values (Fig. 5C). Additionally, elliptical cross-sections explain the subtle difference in RMA regression slopes returned for J, which have strong support in relation to cortical area. An elliptical outline increases the cortical area along one axis compared to a circular outline, but decreases it proportionally along the orthogonal axis as the bone scales up, hence the difference in slope. This is likely further compounded by the highly variable amount of axial flattening described by PC2.

As stated previously, rounded shape has been hypothesised to provide a comparably high safety factor to counteract the considerable bending moments generated when the limb is loaded under torsion during sprawling gate associated with the semi-aquatic habitus (*Blob & Biewener, 1999*; *Meers, 2002*). Mekosuchine specimens diverging from this morphospace may reflect an apparent de-emphasis of torsional loading. The wide spread of the mekosuchine specimens along PC1 is also notable as it appears to display both temporal and geographic correlates. Eocene specimens QM F56058 and QM F56060 span to the positive extreme of PC1. Miocene specimens span between positive and negative values on PC1 along a geographic divide, with Queensland Riversleigh specimens QM F57954 and QM F57955 displaying positive values and the Northern Territory Bullock Creek specimen NTM P907-70 displaying a negative value. The Pleistocene specimen QM F57953 represents the most negative PC1 value. In this latter extreme the shape can be readily related to the development of the enlarged deltopectoral crest of QM F57953. It is possible that this wide spread along PC1 reflects a morphological radiation in the limbs, with an onset in the Oligo–Miocene, related to that observed in the pelvic region (*Stein et*

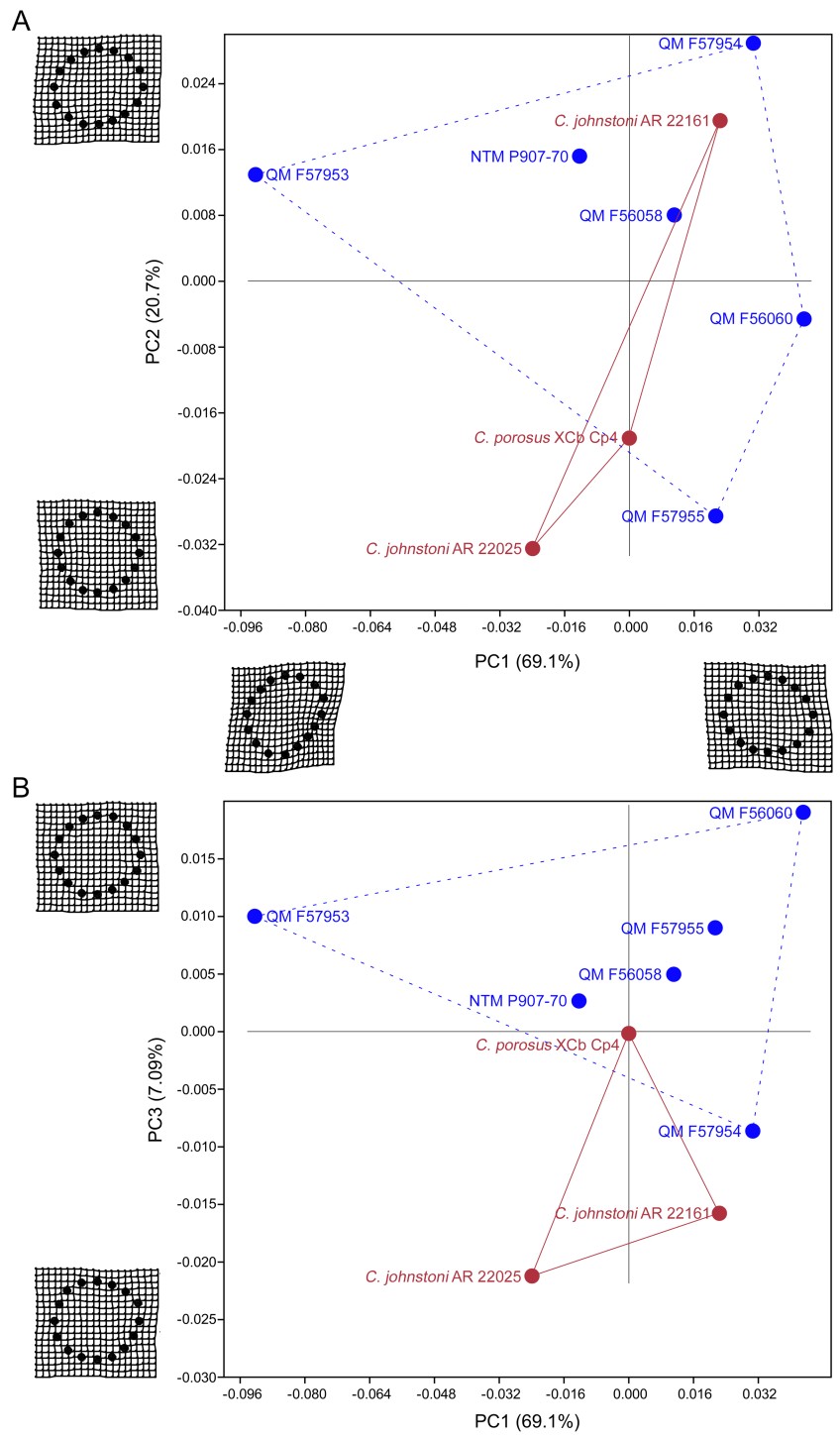

**Figure 4 Results of the principal component analysis.** Principal components analysis of 16 equiangular semi-landmarks collected on the periosteal outline of each 50% cross section. (A) Principal component 1 versus principal component 2, (B) principal component 1 versus principal component 3. The cross sections were extracted from virtual 3D models of the humerus, and the landmark data were corrected for body mass. Freshwater and saltwater crocodiles are shown in red, mekosuchines in blue.

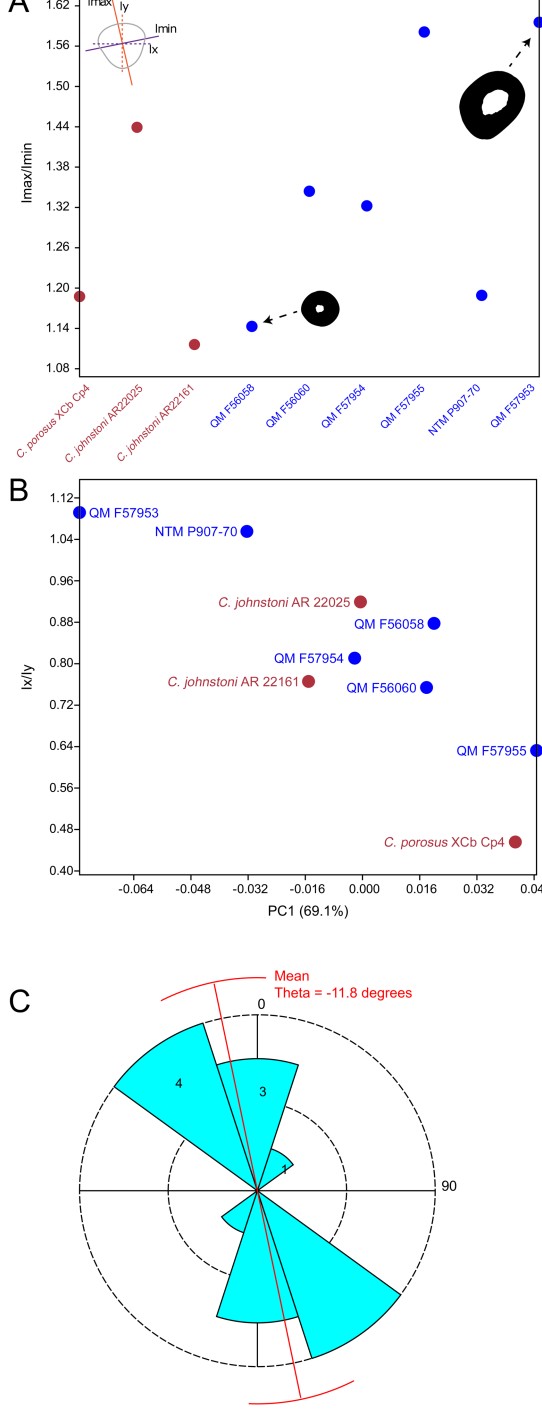

**Figure 5 Plots of cross-sectional properties and shape variance.** (A) Ratios of *Imax* to *Imin* for each specimen, indicating the magnitude of variance in cross-sectional axial symmetry (increased values, greater than 1.0, indicate greater departure from symmetry); (B) principal component 1 (major axis of cross-sectional shape variance in the sample uncorrected for body mass in this case) versus ratios of *Ix* and *Iy*;

**Figure 5 (…continued)**
(C) rose diagram of theta values (in degrees) extracted from cross sections. Theta is the orientation of greatest bending rigidity. Red line indicates average orientation for theta in the sample (−11.8 degrees counter clockwise from 0). Numbers in shaded bin regions correspond to number of samples in bins. Freshwater and saltwater crocodiles are shown in red, mekosuchines in blue.

al., 2017). With so few mekosuchine long bone specimens available, however, this result must be considered preliminary.

It is also clear that a portion of shape variation, i.e., the degree of axial flattening on PC2, does not appear to display phylogenetic correspondence. This is not unexpected because individual variation due to life factors such as nutrition or health that influence the frequency of bone deposition likely introduces variation into the sample (*Wilson & Humphrey, 2015*). An additional caveat to consider is that although the mekosuchine specimens appear to diverge from a morphology associated with semi-aquatic habit, this in itself is not unequivocal proof that the subsequent forms were associated with terrestrial habit. This issue could be resolved if more limb materials became available to establish a larger shape dataset for mekosuchines that could be tested against extinct taxa regarded as terrestrial (e.g., notosuchids) and extant taxa (e.g., *Osteolaemus tetraspis Cope, 1861*, or *Paleosuchus trigonatus Schneider, 1801*) that purportedly show greater aptitude with terrestrial locomotion (*Pol, 2005*; *Eaton et al., 2009*; *Eaton, 2010*; *Pol et al., 2012*; *Bittencourt et al., 2019*). A larger data set would also be necessary to test for any Oligo–Miocene morphological radiation in mekosuchine limb shape and quantify its extent.

The lower Von Mises stresses resulting between mekosuchine and extant crocodile humeral models under finite element analysis and the smaller increase in VM stresses between simulated sprawling and high-walk gaits (Figs. 9A–9B versus 9C, 9E, 9F) might be advantageous in a terrestrial habit, however, if these allowed for the high-walk to be more consistently adopted. It should be noted that extant crocodiles engage in high-walk and even gallop in the case of freshwater crocodiles, so the high stresses engendered in the forelimb do not necessarily preclude these behaviours due to the high safety factors crocodilian limb (*Zug, 1974*; *Blob et al., 2014*). However, the large spike in VM stresses in the saltwater and freshwater crocodile models (Figs. 9A–9B) does potentially explain why a high-walk gait, and galloping in particular, are less commonly employed during locomotion in extant crocodiles, with larger individuals often ceasing to use them (*Zug, 1974*; *Allen et al., 2010*). Lower baseline stresses offer a solution, allowing a larger margin of bodyweight before the high-walk is compromised. Consequently, high-walk gait might be more consistently employed throughout life as bodyweight increases with age.

Only the Riversleigh QM F57954 model (Figs. 6M–6P) displays VM stresses similar to the extant crocodile models (Fig. 9D). This appears to be the result of a large compressive/tensile stress towards the proximal end of the diaphysis, generated by the slight dorsoventral curvature of the diaphyseal shaft (Figs. 6M–6N, 8M–8N). It is possible that the stresses in this model are overestimates, because a degree of allometry has been observed to affect muscle force estimates of either very large or, in this case, very small individuals (*Campione & Evans, 2012*). Smaller individuals also tend to display more gracile diphyseal proportions

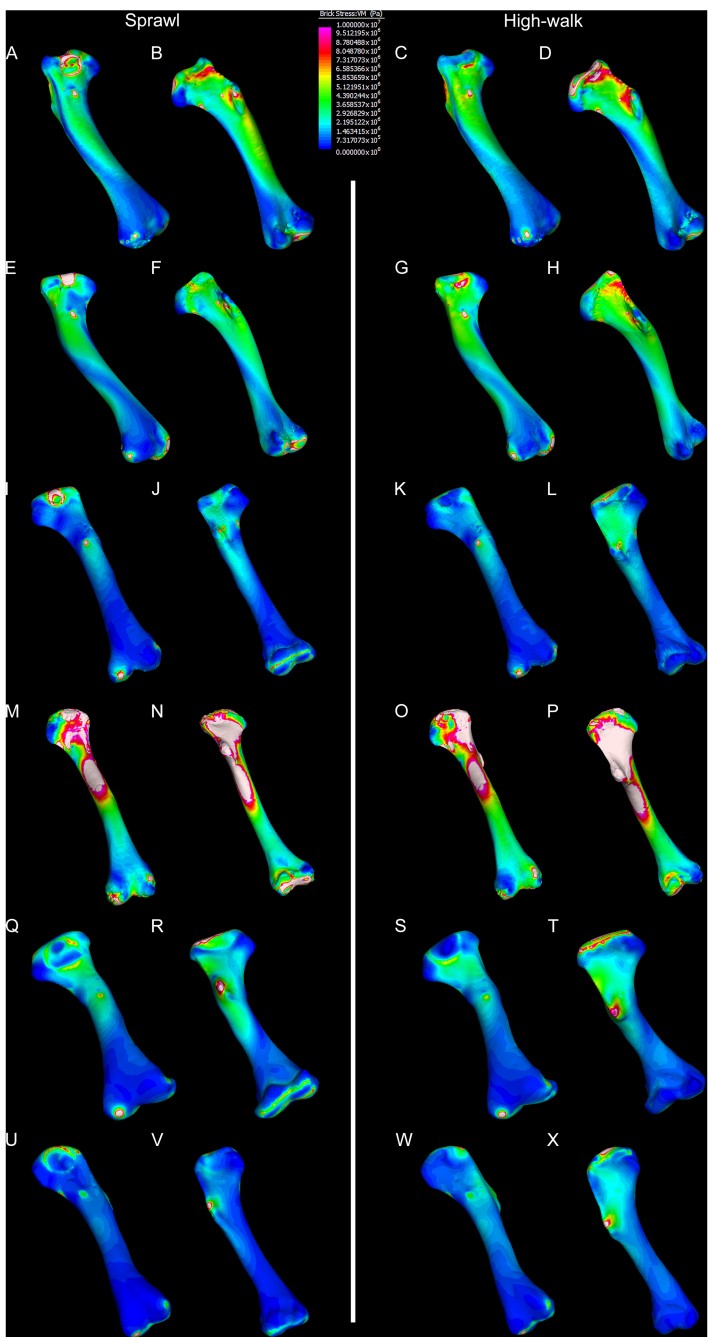

**Figure 6  Results of finite element analysis used to assess how different loading conditions (sprawl, high-walk) affect the patterning of stress engendered by the humerus.** Contour map results of von Mises stresses with output scale chosen to maximise contrast, considering both force of bodyweight due to gravity and musculature. Models are pictured in dorsal and ventral views. Modern *Crocodylus porosus* XCb Cp4 under (A), (B) sprawl and (C), (D) high-walk conditions. Modern *Crocodylus johnstoni* AR 22161 under (E), (F) sprawl and (G), (H), high-walk conditions. Murgon QM F56060 under (I), (J) sprawl and (K), (L) high-walk conditions. Riversleigh QM F57954 under (M), (N) sprawl and (O), (P) high-walk conditions. Bullock Creek NTM P907-70 under (Q), (R) sprawl and (S), (T) high-walk conditions. Floraville QM F57953 under (U), (V) sprawl and (W), (X) high-walk conditions.

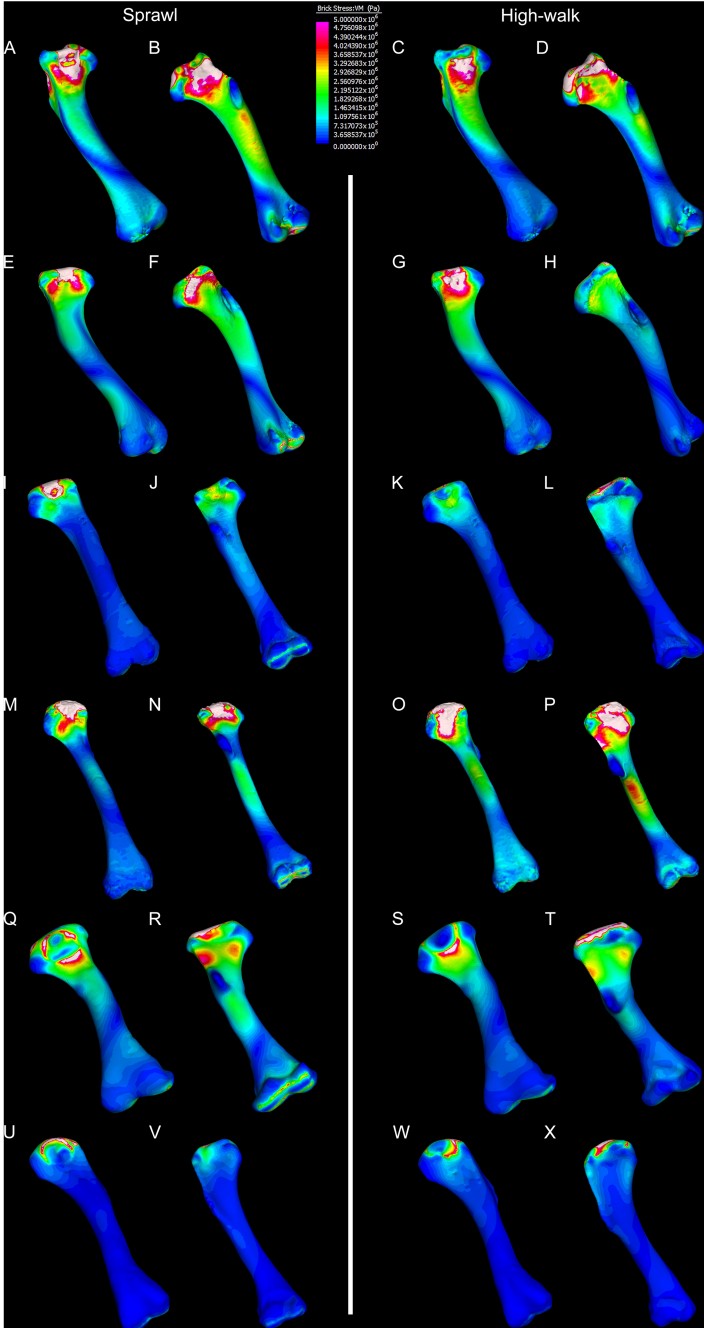

**Figure 7** **Results of finite element analysis used to assess how different loading conditions (sprawl, high-walk) affect the patterning of stress engendered by the humerus under gravity only.** Contour map results of von Mises stresses with output scale chosen to maximise contrast, considering force of body-weight due to gravity solely. Models are pictured in dorsal and ventral views. Modern *Crocodylus porosus* XCb Cp4 under (A), (B) sprawl and (C), (D) high-walk conditions. Modern *Crocodylus johnstoni* AR 22161 under (E), (F) sprawl and (G), (H) high-walk conditions. Murgon QM F56060 under (I), (J) sprawl and (K), (L) high-walk conditions. Riversleigh QM F57954 under (M), (N) sprawl and (O), (P) high-walk conditions. Bullock Creek NTM P907-70 under (Q), (R) sprawl and (S), (T) high-walk conditions. Floraville QM F57953 under (U), (V) Sprawl and (W), (X) high-walk conditions.

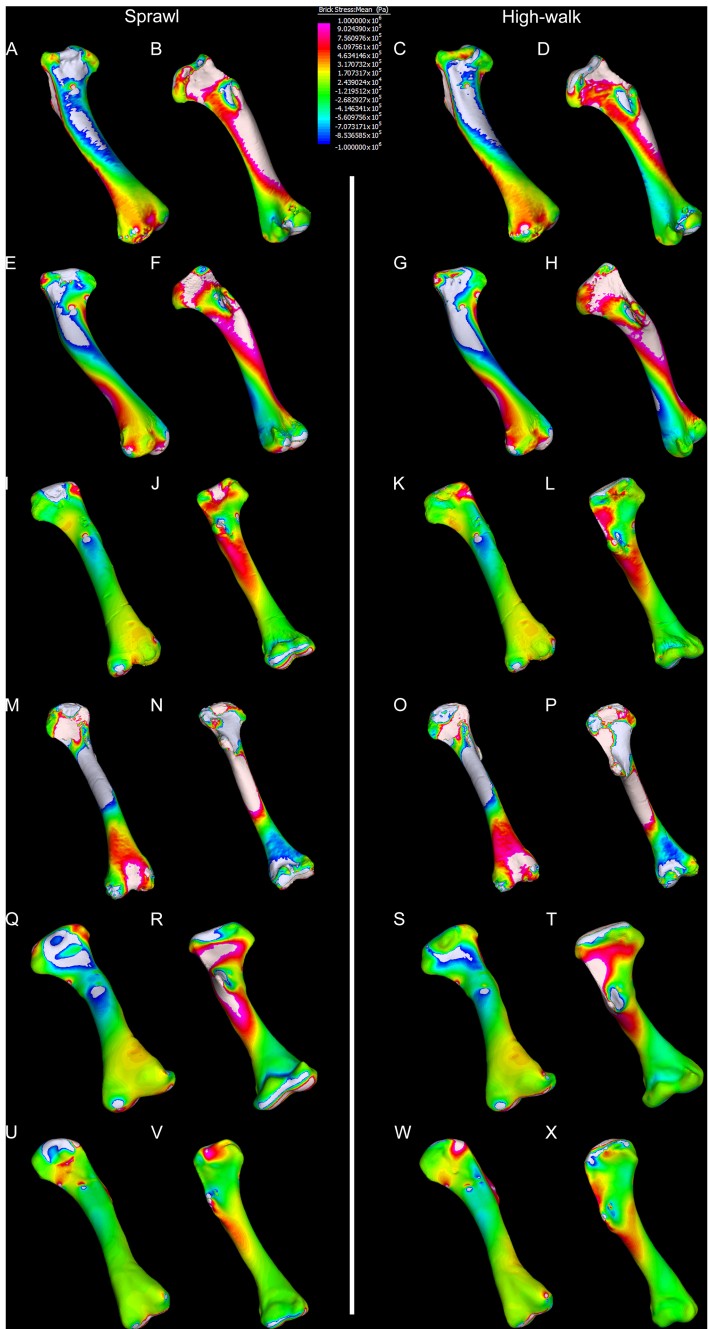

**Figure 8** **Compression and tension results of finite element analysis.** Contour map results of the finite element analysis by mean of the principle stresses with output scale chosen to maximise contrast, considering both force of bodyweight due to gravity and musculature. Yellow to violet indicates the model is in tension, green to blue indicates in compression. Models are pictured in dorsal and ventral views. Modern *Crocodylus porosus* XCb Cp4 under (A), (B) sprawl and (C), (D) high-walk conditions. Modern *Crocodylus johnstoni* AR 22161 under (E), (F) sprawl and (G), (H) high-walk conditions. Murgon QM F56060 under (I), (J) sprawl and (K), (L) high-walk conditions. Riversleigh QM F57954 under (M), (N) sprawl and (O), (P) high-walk conditions. Bullock Creek NTM P907-70 under (Q), (R) sprawl and (S), (T) high-walk conditions. Floraville QM F57953 under (U), (V) sprawl and (W), (X) high-walk conditions.

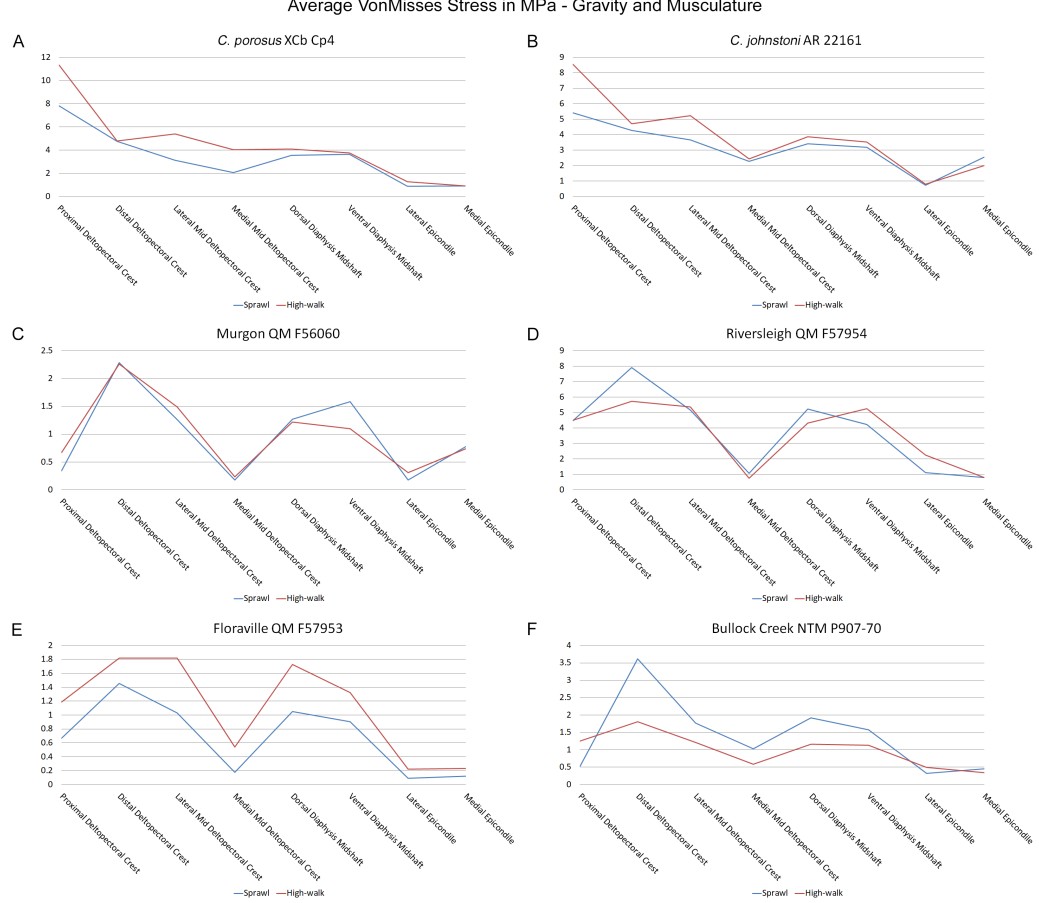

**Figure 9** **Average extracted von Mises stress values.** Average extracted von Mises stresses at measured anatomical locations for each model. (A) *C. porosus* XCb Cp4, (B) *C. johnstoni* AR 22161, (C) Murgon QM F56060, (D) Riversleigh QM 57954, (E) Floraville QM F57953, (F) Bullock Creek NTM P907-70.

(*Iijima & Kubo, 2019*), which may affect the estimation of stresses because the FEA was performed under conditions of isotropy on solid meshes, i.e., not modelling the effect of endosteal shape. Notably, the model generated from NTM P907-70, the Bullock Creek *Baru* taxon, displays lower VM stresses despite being subjected to larger loads (Figs. 6Q–6T, 9F). If the Riversleigh specimens do represent juvenile individuals of a *Baru* species, it is possible that the lower VM stresses of the Bullock Creek NTM P907-70 model represent the effects of allometry both in terms of morphology of the humerus and muscle force.

Both Riversleigh QM F57954 and Bullock Creek NTM P907-70 models are unusual, however, in that peak VM stresses appear to decrease with the shift to high-walk gait at the sampling locations detailed under loading conditions (Figs. 9D and 9F). This is due to net reduction in, alternately, compressive or tensile stresses (Figs. 10D and 10F). This contrasts with the net increase in compressive/tensile stresses that occurs in the modern crocodylian models (Figs. 10A and 10B). The different distributions of compressive/tensile stresses between the models (Fig. 8) suggests a contributing factor may be the presence/absence

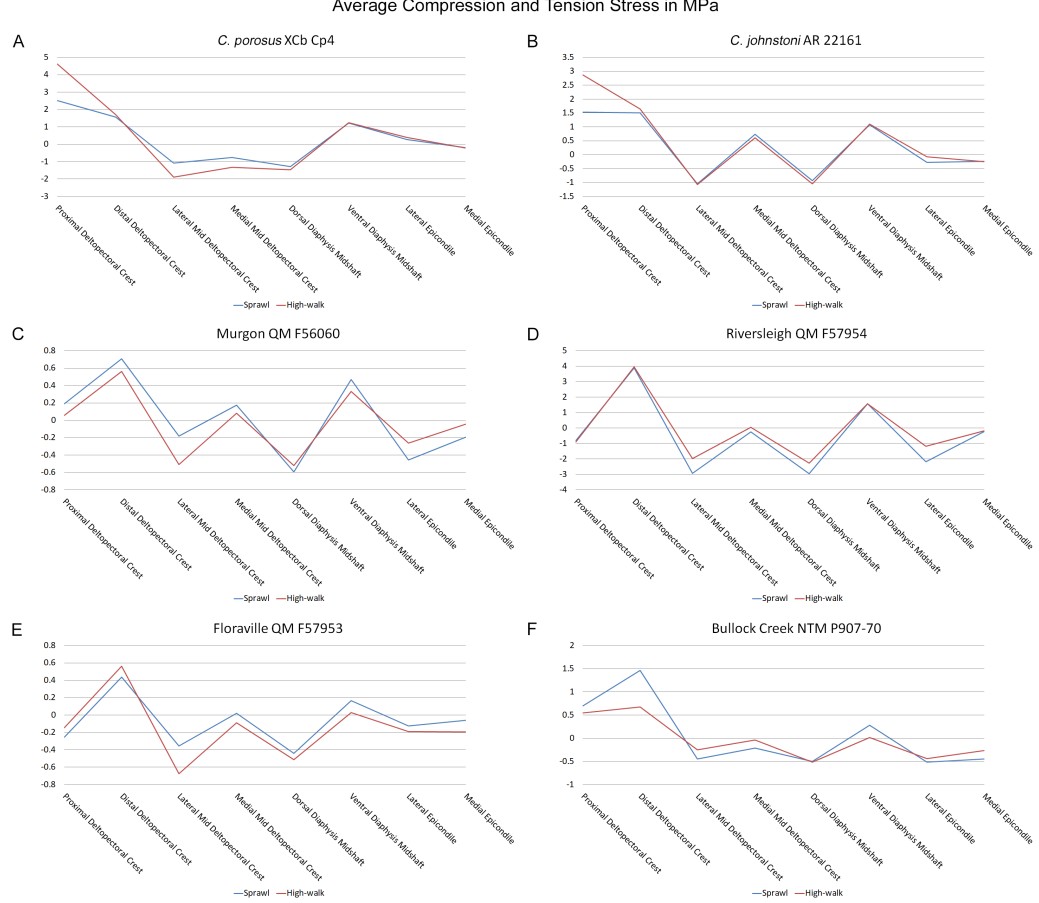

**Figure 10 Average extracted compressive and tensile stress values.** Mean component stresses at measured anatomical locations for each model, where positive values indicate tension and negative values indicate compression. (A) *C. porosus* XCb Cp4, (B) *C. johnstoni* AR 22161, (C) Murgon QM F56060, (D) Riversleigh QM 57954, (E) Floraville QM F57953, (F) Bullock Creek NTM P907-70.

of stresses generated from bending moments (*Blob & Biewener, 1999*; *Blob & Biewener, 2001*). As loads are applied to the models the proximal diaphysis is bent towards the venter and rotated towards the posterior via the deltopectoral crest due to the action of the m. pectoralis. In the highly curved diaphyses of the saltwater and freshwater crocodile models this appears to induce bending moments that generate substantial matching planes of compressive and tensile stress either side of the curve, along the full length of the diaphysis (Figs. 8A–8H). Under high-walk, bending moments further increase compressive stresses on one side of the curve as the full force from bodyweight is applied to the humerus. In response tensile stresses on the other side of the curve increase to match, or at least they hold steady. *Blob & Biewener (1999)* observed a similar result in alligator limbs that were found to be loaded heavily in tension as a result of bending. The only case of a similar matching compressive/tensile stress occurs in the Riversleigh QM F57954 model (Figs. 8M–8P) where the curved diaphysis also appears to induce bending moments, only within a more constrained plane compared to the saltwater and freshwater crocodile models. The

columnar mekosuchine humeral models (Figs. 8I–8L, 8Q–8X) do not appear to induce similar bending moments along their length, generating point stresses instead. These point stresses appear highly variable in extent. This may explain the reduction in peak stresses in the Riversleigh QM F57954 and Bullock Creek NTM P907-70 models, if compressive/tensile stress redistribute unevenly through the diaphysis around the deltopectoral crest with the shift to high-walk.

The variation in structural stresses, between mekosuchine and extant crocodile models, and among the mekosuchine models themselves, appears to illustrate the paradox of curvature in limb bones. Why is there curvature in extant crocodylian limbs at all if it generates such high stresses? Following the predictability hypothesis, an answer may lie in the way structural stresses localise to tightly defined regions in the curved diaphyseal shafts (Fig. 8). This can be traced in the paths of the neutral axis. The neutral axis is strongly bound to the sagittal plane in the saltwater and freshwater crocodile models (Figs. 8A–8H), but runs more transversely in the mekosuchine models, becoming strongly bound to the transverse plane in the Riversleigh QM F57954 model (Figs. 8M–8P). In the saltwater and freshwater models, the result is a stable core that extends along the length of the humerus, with high structural stresses and the concomitant risk of fracture confined to specific planes irrespective of locomotory stance. This would suggest that the ability to maintain a consistent pattern of structural stresses across varied gaits is an important biomechanical advantage of the highly curved diaphyseal shape in extant crocodiles. Structural stresses are distributed more variably throughout the diaphysis of the relatively columnar Murgon QM F56060 (Figs. 6I–6L, 8I–8L) and Floraville QM F57953 (Figs. 6U–6X, 8U–8X) models. Therefore, while a columnar diaphysis may offer advantage in terms of lower baseline stresses, it may trade this off against overall stability across varied gaits.

Does this mean that mekosuchine humeri were more restricted in terms of the gaits they could assume? Likely not, in the same way high-walk is not precluded in a curved humerus thanks to high safety factors. Lower overall structural stresses work to minimise the probability of fracture as much as confining stress regions on the diaphysis. Instead, the differing path of the neutral axis in the mekosuchine models may reflect an alternate biomechanical optimisation. Localisation to the transverse plane suggests axial rather than torsional loadings were predominant, correlating with the divergence from a rounded cross-sectional outline in the mekosuchine specimens. Optimisation for axial loading offers another explanation why peak stresses reduce in the Riversleigh QM F57954 model, following the prebuckled-strut hypothesis, that curvature induces stresses that counteract stresses induced by limb musculature during the step cycle (*Milne, 2016*). The anteriorly bowed diaphysis of the Riversleigh QM F57954 model resembles what has been observed in guanacos (*Lama guanicoe Müller, 1776*). Here the curvature of the diaphysis is theorised to counteract anterior displacement of the proximal end of the humerus induced by the action of the triceps as guanacos engage in habitual climbing in mountainous terrain (*Milne, 2016*). As detailed by *Meers (2003)*, the triceps in crocodiles is differently structured but it is possible that an enlarged triceps might exert a similar anterior displacement during high-walk. If the bowed diaphyses of QM F57954 and QM F57955 represent an adult feature of one of the dwarf Riversleigh taxa rather than a juvenile morphology

that is ontogenically lost, this may indicate that these mekosuchines engaged in similar navigation over rough terrain. This would be advantageous in the Riversleigh mid-Miocene palaeo-environment which is thought to have been composed of a network of closed forests separated by uneven terrain and karst towers (*Archer, Hand & Godthelp, 1994*). This invites consideration that the somewhat discontinuous hydrological history and structuring of the Australian continent, that is more artesian and ephemeral than orogenically influenced (*Habeck-Fardy & Nanson, 2014*), was an important influence on mekosuchine evolution. It must be stated, however, that inferring such a relationship between environment and morphology is speculative at this stage, but could be tested when a larger data set is available to compare a range of crocodilian limb structures with their environments. A potential modern analogue to test this is the South American dwarf crocodile *Osteolaemus tetraspis* which ranges across similarly discontinuous river systems of the West African coast (*Eaton et al., 2009*)

Lastly, there is the case of the Floraville QM F57953 model, which generates the lowest VM stresses (Figs. 6U–6X, 9E). These stresses still conform with values measured in the other mekosuchine models suggesting that the reconstructed mesh generated from Landmark is a valid estimate. The main factor contributing to these low stresses appears to be the substantial enlargement of the deltopectoral crest into a structure that reinforces the diaphysis. This feature is remarkable in its similarity to the humeri of burrowing animals (*Legendre & Botha-Brink, 2018*). The Floraville specimen QM F57953 dates from a period characterised by the onset of aridification in Australia, hence standing bodies of water were becoming rarer (*Martin, 2006*). It is possible that the unusual morphology of this specimen indicates a response to these conditions in the form of expanded burrowing behaviours. This would be consistent with behaviours observed in extant crocodilians: freshwater crocodiles exploit natural recesses in the walls of riverbanks for a consistent thermal environment during the dry seasons in northern Australia, and larger crocodile and alligator taxa utilise and dig out wallows to wait out similar dry spells (*Magnusson & Taylor, 1982*; *Walsh, 1989*; *Mazzotti et al., 2009*). Again, such an ecological inference must be noted as speculation at this stage but this does warrant further research into the digging capacities of crocodylians, possibly comparing a range of extant crocodylians to investigate whether similar morphological features correlate with known digging behaviours.

## CONCLUSIONS

Differences in cross sectional shape appear to be present between available specimens of mekosuchine humeri and the humeri of extant semi-aquatic saltwater and freshwater crocodiles. Saltwater and freshwater crocodiles display rounded cross-sections, whereas mekosuchines appear to vary among a series of elliptical cross-sections. These elliptical cross-sections are also reflected in other biomechanical property measures calculated for the mekosuchine specimens, including *J, Imax/Imin* and theta. This suggests a concomitant variation in locomotory habit, as rounded cross-sectional shape represents an important optimisation for safety-factor to counter the high torsional stresses generated under sprawling gate in semi-aquatic crocodylians. Although these data do not demonstrate

conclusively that mekosuchines were more terrestrial than extant saltwater and freshwater crocodiles, terrestrial habit is also supported by the results of the finite element analysis. Mekosuchine humeral models generate comparatively low structural stresses, which do not spike as greatly when alternating from the sprawl to high-walk gait as stresses in modern saltwater and freshwater crocodile models. Such lower baseline stresses would be an advantage for terrestrial habit as high-walk gait could be more readily employed. However, although structural stresses are higher in the modern freshwater and saltwater crocodile models, their extent is more tightly constrained across varying gaits. Point stresses generated in the mekosuchine models are more variable in their extent. This suggests that the curvature of semi-aquatic saltwater and freshwater crocodile humeri may provide an important biomechanical optimisation for the ability to vary between a range of gaits that trades off higher structural stresses to maintain a core of neutral stress along the whole bone. Transverse orientation of the neutral axis in mekosuchine humeri suggests axial loadings predominated, rather than torsional loadings associated with sprawling gait. Additional morphological variations begin to appear from the Oligo-Miocene onward including the re-emergence of curvature, albeit in a different, more constrained configuration. Novel behaviours can be hypothesised from these developments, including the navigation of un-even terrain and burrowing, but this must be further tested once a larger morphological dataset can be constructed for mekosuchines.

**Institutional abbreviations**

| | |
|---|---|
| **AR** | vertebrate palaeontological research collections, University of New South Wales, Sydney |
| **NTM** | Northern Territory Museum, Alice Springs |
| **QM** | Queensland Museum, Brisbane |

## ACKNOWLEDGEMENTS

The authors thank H Godthelp for his contributions to fossil field work sessions, and J Scanlon and A Gillespie for specimen preparation. We thank A Yates for preparation of and access to specimens held by the Northern Territory Museum. We thank T Hung of the Biological Research Imaging Laboratory, UNSW, Sydney, for micro-CT scanning. We thank A Klinkhamer for CT scan, segmentation and dissection of the saltwater crocodile, used in this study. We also thank C McHenry, M Delfino, R Blob and two additional anonymous reviewers, for review and constructive comments which led to refinement of both analyses and this manuscript. Specimens in Fig. 1 were originally published in (*Stein et al., 2012*) and are reproduced here with permission and due acknowledgement to Alcheringa.

## Funding

This research was funded by Australian Research Council award DE150100862 to Laura Wilson, Australian Research Council grants DP140102656 and DP140102659 to Stephen Wroe, and Australian Research Council grants DP130100197, DP170101420 and DP180100792 to Suzanne Hand and Michael Archer. There was no additional external funding received for this study. The funders had no role in study design, data collection and analysis, decision to publish, or preparation of the manuscript.

## Grant Disclosures

The following grant information was disclosed by the authors:
Australian Research Council: DE150100862, DP140102659, DP130100197, DP170101420, DP180100792.

## Competing Interests

Laura AB Wilson is an Academic Editor for PeerJ.

## Author Contributions

- Michael D. Stein conceived and designed the experiments, performed the experiments, analyzed the data, prepared figures and/or tables, authored or reviewed drafts of the paper, and approved the final draft.
- Suzanne J. Hand, Michael Archer and Stephen Wroe analyzed the data, authored or reviewed drafts of the paper, and approved the final draft.
- Laura A.B. Wilson conceived and designed the experiments, performed the experiments, analyzed the data, prepared figures and/or tables, authored or reviewed drafts of the paper, and approved the final draft.

## Data Availability

STL models and DICOM archives of specimens are available at Morphosource: https://www.morphosource.org/Detail/ProjectDetail/Show/project_id/982.

Raw data spreadsheets and Strand7 files of all analyses are available at Figshare: Stein, Michael; Hand, Suzanne; Archer, Michael; Wroe, Stephen; Wilson, Laura A. B. (2020): Stein et al. 2020 Dataset. figshare. Dataset. https://doi.org/10.6084/m9.figshare.9820961.

Specimens QM F57953, QM F57954, QM F57955, QM F56058 and QM F56060 are available at the Queensland Museum, Brisbane, Australia.

Specimen NTM P907-70 is available at the Museum and Art Gallery of the Northern Territory, Alice Springs, Australia.

Specimens AR 22025 and AR 22161 are available at the University of New South Wales, Sydney, Australia. Specimen XCb Cp4 is deposited with the University of New England, Armidale, Australia.

## Supplemental Information

Supplemental information for this article can be found online at http://dx.doi.org/10.7717/peerj.9349#supplemental-information.

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
