# Peer review of "Quantitatively assessing mekosuchine crocodile locomotion by geometric morphometric and finite element analysis of the forelimb"

_PeerJ, doi:10.7717/peerj.9349_

## Round 0.1 · original submission · Major Revisions

· Academic Editor

Major Revisions

We have 3 reviews with constructive critiques of the MS. There are extensive comments on the science and its presentation. The raw data for the study must be made available. Please detail your changes in a point-by-point Rebuttal to aid re-review of the revised MS, if you choose to resubmit. The reviewers all found some interesting aspects to the study and the fundamental questions are interesting so we hope you do resubmit.

Reviewer 1 ·

Basic reporting

Stein and others analyzed the mechanical properties of humeri in mekosuchine crocodylians that are endemic to the Eocene-Pleistocene of Australia. Mekosuchines are known to exhibit a set of bizarre craniomandibular and postcranial features, and arguably more terrestrial among Crocodylia. Therefore quantitative analysis of mekosuchine appendicular skeleton may give an insight into the osteological adaptation to more terrestrial lifestyle.
References are meticulously cited, background is provided in the introduction, and detailed analysis protocol is given in the materials and methods and in the supplementary data.
The largest concern is the small sample size of extant comparative materials. They only sampled 3 individuals from 2 species (C. porosus and C. johnstoni). There are more 'terrestrial' taxa among living crocodylians (e.g. Paleosuchus and Osteolaemus) that should be included. If mekosuchines are really unique in the limb morphology and mechanical properties cannot be tested here. However, I believe the data presented here may be useful in the future mekosuchine study. PeerJ seems to welcome row data and any scientifically sound analysis.

Experimental design

Experimental design is fine. Authors provides the background info for all the fossil materials. CT scan setting (voxel size, resolution etc.) is in the supplementary but may also be described in the main text.
Authors normalized the mechanical properties with the estimated body mass. However, allometric change in the limb bone shape (Dodson 1975; Meers 2002) still needs to be taken into account. The allometric growths of limb bones are also different interspecifically (Iijima and Kubo 2019). If allometric correction is difficult, authors may want to add caveats by citing above papers.

Validity of the findings

Many of their claims do not have strong support. For example,
1. Differences in the cross sectional shapes between extant crocs and mekosuchines are unclear. As I see Fig. 4, I cannot find any trends. In Fig. 5, Imax/Imin is highly variable among 3 extant individuals and among mekosuchines.
2. Similarly, there is no clear difference in the stress distribution patterns between extant crocs and mekosuchines (Fig. 6). Patterns in the Murgon specimen (Fig. 6I-L) look similar to those in extant crocs (Fig. 6A-H).
3. "Deviation of the mekosuchine specimens from rounded cross-sectional shape", and "apparent de-emphasis of torsional loading", which are repeatedly mentioned in the abstract and discussion (lines 515-518) do not have strong support. Comparisons with extant samples are not enough, and differences between extant crocs and mekosuchines are ambiguous.
These issues can be addressed by qualifying the statements throughout the abstract and the main text.

Additional comments

Materials & methods-Analysis of cross-sectional shape and biomechanical properties: Explanation of J (polar second moment of area) seems to be missing.
Lines 345-346 'Small values for Imax/Imin indicate bending strength along Imin and larger values indicate greater bending strength along Imax (Ruff, 1987)': Smaller Imax/Imin indicate similar Imax and Imin, and larger values indicate discrepancy in Imax and Imin?
Line 406: primary actors: primary protractors?
Lines 464-466: Please provide slope values in your study and Meer (2002) for comparison.
Lines 472-476: Is PC3 really related to cross-sectional flattness?
Line 496: It should be more informative if the difference of stress magnitude is mentioned as percentages.
Lines 551-562: Similarity of Riversleigh mekosuchines and extant crocs in the stress patterns might be due to small size in Riversleigh forms. Smaller individuals generally have very slender limb bones.
Lines 568-571: 'This result confirms that the morphologies of mekosuchine humeri generate a different loading regime than that which occurs in extant crocodylians': Replace confirm with 'may suggest' or sorts.

·

Basic reporting

In general the language is well presented, though I have several editorial suggestions (detailed in section 4). The Introduction provides useful background, and the delineation of sections follows standard presentation. The figures are generally executed well, though I do have suggestions about how to refer to them more to improve their effectiveness (detailed below). Some raw data (e.g. PC eigenvalues and vectors) are not clearly supplied at this point.

Experimental design

The research is original and within the scope of the journal. The research gap that this work addresses is noted clearly. Technical execution of the work is strong, with methods described in detail and following ethical standards.

Validity of the findings

I have a number of points to raise in the context of the findings, in order of importance:

(1) L453-466. I am struggling with the regressions of cross-sectional shape reported here and illustrated in Fig. 3. First, it is not clear whether the salt and freshwater crocodiles plotted in these figures are also included in the sample that is used to calculate the reported slopes. If these slopes are supposed to reflect mekosuchine traits, they really should not have those modern taxa included. Second, there are no confidence intervals reported for these slopes. This makes comparison to the slopes reported by Meers quite challenging. The assessment provided, that they are typically similar, includes a span of differences (as best as I can determine) of from ~0.04 to well over 1.0. A direct notation of which numeric slope reported from Meers should be related to which slope from the present study would be very helpful – it is not exactly explicit as currently reported in the text, without referring out to the Meers paper, which preferably would not be necessary. I will add that it would be helpful to potentially compare the slopes reported to predictions for isometry or allometry. I also don’t recall seeing the mode of regression being reported (reduced major axis would be most appropriate). But overriding all of these points is that once restricted to a mekosuchine sample, the number of samples is very small, which will likely make confidence intervals larger than will be straightforward to compare to other patterns. Overall, I would encourage the authors to rethink how this comparison is performed.

(2) L468-481; 507-510. I also struggle with the presentation of the principle components analysis. This presentation is different than I am familiar with from other studies, as it appears to use the coordinates of each of the landmark points as an original variable, without correcting for size. There are no data reported that I can find on the eigenvalues/eigenvectors/loadings of the original variables on the new axes. Moreover, interpretation is quite challenging, as the mekosuchines appear to san nearly the full extent of morphospace. I do see the different shapes represented on the extremes of the PC axes, but with the mekosuchines spanning most of these dimensions it is difficult to resolve what can be inferred with confidence about implications for the functional abilities of the group. It is only in the Discussion that the point is made about the specimens plotting with respect to time and geography. However, the groups of two on each side of PC 1 are not distinct with regard to geography (Queensland specimens on each extreme) or time (Miocene specimens on each extreme). I think the presentation of this analysis also needs reconsideration.

(3) L488-497. The FEA results are one of the strongest points of the paper, with the comparisons of the different replicated loading conditions providing interesting insights, such as that previous experimental findings of increasing stresses with upright posture may have been present in ancient crocodilians as well. (I will note here that, the point made in the Discussion L564-566, about two specimens showing a decrease in stress with upright posture, was not very evident in the presentation of these data in the Results). This presentation of the results is very efficient, but I wonder if it might be over-distilled. Readers could probably use a bit more guidance as to what they should be gleaning distinctly from the four visually intense figures that are referenced in this paragraph.

(4) L515-517. The idea that a shift to non-circular cross sections might reflect a shift away from torsional loading is reasonable – Blob 2001 (Evolution of hindlimb posture in non-mammalian therapsids: biomechanical tests of paleontological hypotheses. Paleobiology 27:14-38) makes the same inference for non-mammalian therapsids. However, I think the analyses that produced the shape assessment might be reconsidered in order to make the case for this argument in a more straightforward fashion.

(5) L520-526. I would be very cautious about inferring any evolutionary trajectories based on the limited sample of specimens analyzed.

(6) L594-608. Some of the speculations offered here are stimulating, but I think they may stretch a little too far. With only 4 specimens analyzed, for example, I have a hard time with the statement that “the somewhat discontinuous hydrological history and structuring of the Australian continent, that is more artesian and ephemeral than orogenically influenced (Habeck-Fardy & Nanson, 2014), was an important influence on mekosuchine evolution.” In one sense, it probably was in a lot of ways. But to infer it specifically for humeral shape, I’m not sure there is enough evidence to really make such a statement. I’d recommend re-looking at this whole section, this latter point in particular, with a finer comb to evaluate how far it is appropriate to propose some of the ideas presented.

(7) L634-635. Here again, this statement about the likelihood of facultative burrowing seems too strong.

Additional comments

L48. I think “constraint” is the word that should be used here, rather than “restraint”. But actually, I don’t think this broad statement is really needed for the abstract, which typically focuses on the study conducted. So, I’d remove the whole first sentence, and limit the generalization to ”Morphological shifts observed in the fossil record of a lineage might indicate concomitant shifts in ecology of that lineage.”

L70. I’d recommend inserting “often” before “is reflected”, since there are many exceptions to such expectations.

L98. To clarify the reference in the sentence, I’d change “this” to “such terrestrial specialization”.

L100. “verses” should be “versus”.

L103. “a held near” should be “are held nearly”.

L128-129. For clarity, I would change “Variation in curvature may not necessarily relate to strain either; curvature has been shown to allow physical space for muscle belly packing” to “ Variation in curvature also might relate to factors other than strain, such as allowing physical space for muscle belly packing.”

L134. Change “is” to “are”.

L137. “short time span” – how short? If referring to the whole duration of mekosuchines, isn’t it over several tens of millions of years?

L144. In fact, torsion has also recently been reported as a primarly loading mode for the humerus of crocodilians: Blob et al 2014, Integr Comp Biol 54:1058–1071.

L149. I’d add a comma after “plane”; also, it might be helpful to add a citation of Reilly & Blob 2003 (J Exp Biol 206:4327-4340) to the Blob and Biewener 1999 citation, since it verified the finding of increases in loading with upright posture and explained its mechanism.

L152. Move “from extant crocodylian humeri” before “indicative” for clarity.

L155. Add something like “with trophic habits” after “correlate”, to show what they correlate with.

L178. I’d add “the loading consequences of “ before “cross-sectional shape variance”.

L185. Add a comma after ‘mitigation’ – but also specify ‘same imperatives of stress mitigation’ as what/who?

L189. Add a comma after mekosuchines. Also, in this context of interpreting what deviations from rounded shape might imply for the presence or reduction of torsional loading, a useful reference for comparison would be Blob 2001 (noted above).

L229. Change “boarder” to “border”.

L268-272. I am not following the text in this section. It is fine to acknowledge that phylogenetic relatedness was not considered in this study, but I am not sure how it can be justified by saying that not considering phylogeny prevents morphology of interest from being exluded.

L402, etc. Are there any experimental data, such as EMGs, that can provide a basis for these reconstructions of muscle activity, beyond the anatomical analysis from Meers 2003?

L414. Hyphenate “pulley-like”.

L514. “Considerable moment forces”. This phrasing is confusing. Forces are part of what determine moments. Is “forces and moments” what is meant? Also, things aren’t really ‘loaded under torsional stresses’ – the stresses result from experience a twisting load. Probably just ‘loaded under torsion’ is better.

L574. To be most accurate, it would be best to add “as a result of bending” after “tension”. In addition, the rest of the text in this paragraph, indicating increases in tension relating to increases in compression, is a bit hard to follow and would benefit from some reworking.

L598. “represents” should be “represent”.

L622. These behaviors might not be precluded due to the high safety factors of crocodilian forelimb bones, which could be referenced here (e.g. Blob et al. 2014).

Reviewer 3 ·

Basic reporting

A1: The manuscript was generally well-written but there were certain sections that were over-generalized. For instance, line 70 includes “offer niche opportunities for those that adapt” but phenotypic plasticity could lead to morphologies that could also make use of those niche opportunities. Lines 143 – 144 suggest that the loading regime would be similar between the humerus and femur in the alligator yet Blob et al. 2014 (https://academic.oup.com/icb/article/54/6/1058/636968) found that the safety factors and yield strains are different. My guess is that the text was supposed to read that torsional loads would be important in the humerus since it is also important in the femur.

A2: Given how bone curvature plays such a substantial role in their study, it is surprising that it is not mentioned in the abstract. The effects of bone curvature on limb bone stresses is an important component to evaluating to the locomotor biomechanics of tetrapods and played a particularly valuable role in evaluating the morphological differences between their taxa, so it would be useful to incorporate some of that information in the abstract.

A3: The raw data do not seem to be shared, other than the data presented in Table 1 and the scan settings in the Supplementary Material.

Experimental design

B1: While the text is generally well-written, additional information about the geometric morphometrics would help clarify some details. For instance, the landmarks were described to have been obtained from the 3D models by extracting the Cartesian coordinates where 16 equidistant radii extended to the edges of the bone. Based on this description, these points seem to be more traditionally defined as semi-landmarks in the geometric morphometrics vernacular. To what extend were these landmarks homologous across the taxa? Addressing this information as well as identifying the program (and potentially functions) used to run these analyses would help clarify this section.

Validity of the findings

C1: The manuscript could be strengthened by addressing any validation models or sensitivity analyses on the finite element analyses (e.g., see Porro et al. 2013 and references therein: https://onlinelibrary.wiley.com/doi/full/10.1111/joa.12080). The material properties assigned to the FEMs were reported but those values appear to be based on mammals or sharks and most (if not all) on cranial material based on the references provided. Currey 1988 (https://www.sciencedirect.com/science/article/pii/0021929088900061) reports that the Young’s modulus is about 12 GPa in the alligator femur is about 6.65 GPa in the alligator cranium (see references in: https://onlinelibrary.wiley.com/doi/pdf/10.1002/jmor.10627), so choosing values that better matched the stylopods of crocodilians would seem to be more biologically relevant. Similarly, it is unclear whether FEA output generally remains robust despite modelling the lattices with properties of steel (perhaps a reference could be included to address this if that topic has already been discussed elsewhere). The Zapata et al. 2009 (“Material properties of mandibular cortical bone in the American alligator” paper in the Bone journal) could help support the value chosen for the Poisson ratio. Although validation models may not necessarily be part of this manuscript, it could be addressed by relating the results to published work on alligator stylopods (e.g., Blob et al. 2014 reports strain data for the alligator humerus during terrestrial locomotion: https://academic.oup.com/icb/article/54/6/1058/636968).The manuscript compares the outputs between the mekosuchine and extant crocodilian models, but it would also be useful to compare the output from the extant crocodilian FEMs with empirical data on crocodilians.

C2: On line 504, PC1 is suggested to illustrate “a radiation in mekosuchine locomotory habitus” but that is not clear from the plot given that the mekosuchines are spread throughout the morphospace. What seems clear is that only the Crocodylus occupy the lower left square and the Pleisocene mekosuchine occupies the top left corner of the PCA plots. What is reported in lines 508 – 510 sounds correct but it’s unclear how that necessarily suggests a radiation.

C3: Also, the differences in the extreme ends of PC2 and PC3 are fairly subtle based on the thin-plate splines provided, so it would be useful to report the variables that loaded strongly on these axes so it is clearer how the factors were interpreted.

Additional comments

The present manuscript integrates geometric morphometrics and finite element analysis to evaluate the locomotor capabilities of mekosuchine crocodylians compared to extant taxa, based on the functional morphology of the humerus. Specifically, morphometrics of the cross-sectional geometry and whole-bone finite element models are used to evaluate the loading regimes that could have been supported by mekosuchines. While the introduction sets the stage to evaluate how variation in bone morphology (especially curvature) could affect the moment arms and, thus, bone stresses associated with terrestrial locomotion, the current draft of the manuscript could be improved with further clarification (see sections above). Not only would additional justifying information help to strengthen some of the conclusions drawn from the data (i.e., be less speculative) but would also help highlight some of the important details that may be overlooked at the moment.

D1: Why study the forelimb when hind limbs tend to be primary propulsor in reptiles and have been the focus of most studies in crocodilian locomotion? There are some really valuable studies on different limb function but it wasn’t clear whether that was part of the motivation of this study.

D2: It is great that the authors have included overviews of the bone stress data across individual bones but the current presentation of the data may not emphasize the patterns enough. Perhaps changing the format such that the y-axis remains as the stress values but that the x-axis is the distance (or landmark) along the bone, with each taxon being plotted as a line graph rather than bar graphs. For instance, see some of the graphs in Doube et al. 2018 (https://royalsocietypublishing.org/doi/full/10.1098/rsos.180152)

D3: Figure 5C doesn’t seem necessary. If the point is to emphasize that there was variation in theta values, that could be expressed by providing the raw values in the main text.

---

## Round 0.2 · Minor Revisions

· Academic Editor

Minor Revisions

Well done with the revisions. All 3 reviewers are pleased and 2 just have minor comments; 1 has none further. The main comment is to clarify the variation/lack thereof along PCs2+3. No re-review will be needed- and take your time with revisions, I know it is a difficult time for many.

Reviewer 1 ·

Basic reporting

I have previously reviewed this manuscript. Authors addressed all the points raised by three reviewers and I am happy with the revised MS.

Experimental design

no comment

Validity of the findings

no comment

·

Basic reporting

The editing performed for this revision has done a good job of clarifying requested topics, and appropriately qualifying some of the proposed ideas. Additional references to figures are appreciated.

Experimental design

The additional descriptions added to the Methods are helpful and reinforce that the execution of the analyses was appropriate.

Validity of the findings

Adjustments have been made to several analyses (e.g. regressions and slope comparisons) that give appropriate confidence in the findings presented.

Additional comments

Overall the revisions have been quite thorough in addressing all the points raised in review. The authors have done a good job incorporating some additional context for many of the issues noted in the reviews. I really only found a few further typographical items to mention.

L89. Add a comma between “behaviours” and “and”.

L119. Change “curvatures” to “curvature”.

L165. I would use either “second moment of area” rather than “second moment area of inertia.”

L378. Use “nearly” in place of “near”.

L495. “Extend” should be “extends”.

Reviewer 3 ·

Basic reporting

The authors mention: “The raw data has been set to open viewing on Figshare prior to resubmission (DOI: 10.6084/m9.figshare.9820961).” The link does not seem to reference the correct data set on FigShare, though. Is this the correct link? https://figshare.com/articles/Stein_et_al_2020_Dataset/9820961.

Line 348: Holt and Trinkaus 2006 seems to be missing from the list of references.

Experimental design

The experimental design and research objectives are well described.

Validity of the findings

Line 480: how strong is this correlation? Provide the correlation coefficient to support this claim or change to “positively correlates”. Table 3 contains very useful information about the regressions, but mentioning a strong correlation implies that a correlation was calculated from a statistical standpoint.

Please report the PCA loadings in the Supplementary Information to make it more accessible, as the differences being described in the results are subtle across PC 2 and almost non-existent on PC 3. Making broad generalizations about the patterns across these axes is a bit spurious without the quantitative information to support these statements.

Additional comments

The revised manuscript is much improved and better highlights the impact that the results provide in interpreting the locomotor capabilities of mekosuchines while paying closer attention to which hypotheses could be tested with a larger and more comparative data set. I have some additional recommends to help strengthen the manuscript further but am pleased with the quality of the manuscript overall.

---

## Round 0.3 · Minor Revisions

· Academic Editor

Minor Revisions

Thanks for your attentive revisions, which I have checked and am pleased with- the MS has improved nicely from peer review. It is indeed an interesting paper. You've also done a great job including a wide array of data from the study to make it maximally open.

I request that you provide the microCT data with the STLs on MorphoSource as those are the fundamental raw data for the study.

Otherwise I am ready to recommend acceptance. Thank you!

---

## Round 0.4 · accepted · Accept

· Academic Editor

Accept

Thank you! The paper is in fine shape and I am happy to accept it.